health and disease and epidemiology/
mathematical modelling/applied mathematics

effective reproduction number, mathematical models, meta-population models, COVID-19

**Author for correspondence:**
D. C. P. Jorge
e-mail: danielcpjorge98@gmail.com

6167795.

# Estimating the effective reproduction number for heterogeneous models using incidence data

D. C. P. Jorge[1,3], J. F. Oliveira[2], J. G. V. Miranda[3], R. F. S. Andrade[2,3] and S. T. R. Pinho[3]

[1]Instituto de Física Teórica, Universidade Estadual Paulista—UNESP, R. Dr. Teobaldo Ferraz 271, São Paulo 01140-070, Brazil
[2]Center of Data and Knowledge Integration for Health (CIDACS), Instituto Gonçalo Moniz, Fundação Oswaldo Cruz, Salvador, Bahia, Brazil
[3]Instituto de Física, Universidade Federal da Bahia, Salvador, Bahia, Brazil

(iD) DCPJ, 0000-0003-4707-3234

The effective reproduction number, $\mathcal{R}(t)$, plays a key role in the study of infectious diseases, indicating the current average number of new infections caused by an infected individual in an epidemic process. Estimation methods for the time evolution of $\mathcal{R}(t)$, using incidence data, rely on the generation interval distribution, $g(\tau)$, which is usually obtained from empirical data or theoretical studies using simple epidemic models. However, for systems that present heterogeneity, either on the host population or in the expression of the disease, there is a lack of data and of a suitable general methodology to obtain $g(\tau)$. In this work, we use mathematical models to bridge this gap. We present a general methodology for obtaining explicit expressions of the reproduction numbers and the generation interval distributions, within and between model sub-compartments provided by an arbitrary compartmental model. Additionally, we present the appropriate expressions to evaluate those reproduction numbers using incidence data. To highlight the relevance of such methodology, we apply it to the spread of COVID-19 in municipalities of the state of Rio de Janeiro, Brazil. Using two meta-population models, we estimate the reproduction numbers and the contributions of each municipality in the generation of cases in all others.

## 1. Introduction

The human population gets along with different microorganisms. Some of them lead to transmitted diseases that result in

epidemics, even pandemics such as SARS-CoV-2. It is very important to define reliable measures to characterize the spread of those pathogens, both at the beginning and in the course of epidemics.

The reproduction number $\mathcal{R}(t)$ indicates the average number of new infections resulting from a single infected individual at any time $t$. When it is greater than 1, each infected individual tends to generate more than one infected individual, leading to an epidemic outbreak. At the start of the epidemic, when $t = 0$ and it is assumed when all of the population is susceptible, $\mathcal{R}(t)$ is referred to as the basic reproduction number $\mathcal{R}_0$. Due to the recovery of infected individuals, a fraction of the population becomes immune to the disease, acting as a barrier to the transmission of the disease and creating conditions for the emergence of herd immunity. Therefore, $\mathcal{R}(t)$ takes into account the evolution of the number of susceptible individuals in the population and is usually referred to as the effective reproduction number. Actually, it is a much relevant metric to characterize the current level of the disease propagation.

To study reproduction number, the renewal equation is often used alongside incidence data, within an approach based on the generation interval distribution popularized in [1]. Several works indicate how to obtain this distribution using empirical data by tracking the case-to-case transmission process taking into account statistical methods [2–6]. However, estimating such distribution relies on epidemiological and empirical studies that, for several systems, are very difficult to analyse or even do not have the required data for the envisaged evaluation. An alternative is to obtain an analytical form of this distribution using models or assumptions on the epidemic process [1,7–11]. Yet, these methods usually rely on very simple assumptions or models, such as the SIR and SEIR, which assume that there is only one type of infectious infected individuals. Thus, systems that present heterogeneity, either in the host population or in the expression of the disease, usually do not satisfy all conditions required for the analytical approach for the generation interval distribution. In this work, we focus on establishing a method to obtain an analytical form for the generation interval distribution and reproduction number for an arbitrary heterogeneous compartment model. The multiplicity of infected compartments naturally allows the reproduction number and generation interval distribution to be expressed in terms of contributions related to each infected compartment. In fact, our method leads to a matrix of reproduction numbers that establishes how each infected compartment is able to generate new infections on the other ones. For some models, such as the SEIR, this matrix can be reduced to one reproduction number, and consequently, one generation interval distribution. To illustrate the methodology, we apply the method to a meta-population model to analyse the role of spatial heterogeneity in the spreading of COVID-19 in the Southeast of Brazil.

Our work is organized as follow: in §2, we introduce a very general heterogeneous compartmental model that can be reduced to most models in the literature, and use it to develop our general methodology, including expressions for the reproduction numbers that also allow to express it in terms of actual data. Then, in §3, we apply the method to a simple meta-population model with the commuter flow of individuals. In §4, we use actual data of the emerging SARS-CoV-2 coronavirus pandemic in the municipalities belonging to the metropolitan region around Rio de Janeiro, Brazil, to estimate the reproduction numbers that emerge from the system. Additionally, we reconstruct the time series of cases from the contributions of each municipality in the propagation of the disease.

# 2. Reproduction number in a heterogeneous population

## 2.1. A general infection-age model

In a heterogeneous population, individuals can be discernible by different traits such as age, spatial locations, behaviour, different susceptibility to diseases or any other factor that may distinguish them from each other. In this section, we consider those different traits of the population using a heterogeneous compartmental model. This modelling approach separates the individuals in $m$ homogeneous compartments that represent the different traits of the population. For the purpose of evaluating the reproduction number, it is necessary to consider only the subset of variables encompassing the infected compartments, i.e. those compartments containing individuals that are able to transmit the etiologic agent to uninfected individuals. This way, we look at the infected individuals of the population and gather them in the compartments that correspond to their traits. We also let $m = n + m_0$, where $n$ and $m_0$ indicate the number of infected and non-infected compartments, and consider only the first $n$ compartments.

In order to access information regarding these compartments, we define the vector $x(t, \tau) = (x_1(t, \tau), \ldots, x_n(t, \tau))$, where $t$ and $\tau$ indicate, respectively, the calendar time and the infection-age, i.e. the time elapsed since an individual got the infection. The elements of $x$ can be interpreted as the infection-age distributions such that, for each calendar time $t$, $x_i(t, \tau) \, d\tau$ gives the number of individuals in compartment $i$ with infection-age $\tau$. Of course the condition $x(t, \tau) = 0$ for $\tau < 0$ must be satisfied. The total number of individuals in each compartment, denoted by $X(t)$, can be obtained by integrating $x(t, \tau)$ with respect to $\tau$ from zero to infinity

$$X(t) = \int_0^\infty x(t, \tau) \, d\tau,$$

where $X(t) = (X_1(t), \ldots, X_n(t))$. For the sake of simplicity, we use vector and matrix notation in the equations, whereby they are identified by bold font and their elements are indicated by subscripts $M(t, \tau) = [M_{ij}(t, \tau)]$. As usual, integrals and derivatives with respect of scalar variables operate component wise.

Drawing a parallel with van den Driessche's next generation method [12], we re-write the $n$ compartment equations using two vectors. So we define $\mathcal{F}(t) = (\mathcal{F}(t), \ldots, \mathcal{F}_n(t))$ as the rate of appearance of new infections at each calendar time, and $\mathcal{V}(t, \tau) = (\mathcal{V}_1(t, \tau), \ldots, \mathcal{V}_n(t, \tau))$ as the rate of transfer between compartments for a given $\tau$. Therefore, $\mathcal{F}(t)$ describes the flow from non-infected compartments into infected ones and depends on $X(t)$. On the other hand, $\mathcal{V}(t, \tau)$ is related to the flow between infected compartments, such as going from one stage of the disease to another, or from infected to non-infected ones, such as recovery. $\mathcal{V}(t, \tau)$ must depend on $x(t, \tau)$. Thus, a usual infection-age model can be written as

$$\left( \frac{\partial}{\partial t} + \frac{\partial}{\partial \tau} \right) x(t, \tau) = -\mathcal{V}(t, \tau) \tag{2.1}$$

and

$$x(t, \tau = 0) = \mathcal{F}(t). \tag{2.2}$$

An infection-age model is a set of partial differential equations (PDE). Most infectious disease models are usually expressed in the ordinary differential equation form (ODE). However, they can also be written as a infection-age PDE system. In fact, the Kermack–McKendrick SIR and SEIR models [13] are the special cases of their infection-age counterparts. Thus, if we integrate (2.1) from zero to infinity with respect to $\tau$, the PDE model in $x(t, \tau)$ is converted into a $X(t)$ ODE model

$$\frac{d}{dt} X(t) = \mathcal{F}(t) - \int_0^\infty \mathcal{V}(t, \tau) \, d\tau. \tag{2.3}$$

Though a general infection-age model is the starting point of our methodology, it can be extended to ODE systems by simply identifying the $\mathcal{F}$ and $\mathcal{V}$ terms in the model equations. We demonstrate this for usual ODE models in electronic supplementary material, S1.

Moving forward, we solve equation (2.1) by integrating along the characteristic lines which, as one concludes from the l.h.s. of (2.1), are lines of slope 1, i.e. $t = \tau + c$, where $c$ is an arbitrary constant. Fixing a point $(t_0, \tau_0)$ and introducing a new variable $\omega$, we find that $u_i(\omega) = x_i(t_0 + \omega, \tau_0 + \omega)$ are functions that provide the values of the compartments along the characteristic lines. After straightforward calculations [14], we obtain

$$\frac{d}{d\omega} u(\omega) = -\overline{\mathcal{V}}(\omega), \tag{2.4}$$

where $\overline{\mathcal{V}}_i(\omega) = \mathcal{V}_i(t_0 + \omega, \tau_0 + \omega)$. In epidemic models, $\overline{\mathcal{V}}_i(\omega)$ are usually linear equations, so that the resulting system becomes

$$\frac{d}{d\omega} u = -\frac{\partial \overline{\mathcal{V}}}{\partial u} u, \tag{2.5}$$

where $-\frac{\partial \overline{\mathcal{V}}}{\partial u} = [-\frac{\partial}{\partial u_j} \overline{\mathcal{V}}_i(\omega)]$ is the matrix that defines the linear system. Assuming that there are no infected individuals prior to $t = 0$, we only need to take into account its solution for $t > \tau$, leading to $\omega = \tau$, $t = \tau + t_0$ and $\tau_0 = 0$. The solution for a linear system can be written as

$$u(\omega) = \overline{\Gamma}(\omega) \, u(0), \tag{2.6}$$

where $\overline{\Gamma}(\omega) = \Gamma(t, \tau)$, is the fundamental matrix obtained by solving (2.4). Therefore, identifying $\mathcal{F}(t_0) = u(0)$ in (2.2),

$$x(t, \tau) = \Gamma(t, \tau)\,\mathcal{F}(t - \tau). \tag{2.7}$$

For linear $\mathcal{V}(t, \tau)$'s, which is the case of all models considered henceforth, the $\Gamma(\tau)$ components are exponential functions. Nevertheless, equation (2.7) can also be used to express the solution of a general nonlinear $\mathcal{V}(t, \tau)$.

Our next assumption is to consider the following ansatz for $\mathcal{F}(t)$:

$$\mathcal{F}_i(t) = \sum_j^n \int_0^\infty \Omega_{ij}(t, \tau)\,x_j(t, \tau)\,\mathrm{d}\tau, \tag{2.8}$$

whereby $\Omega(t, \tau)$ is a matrix whose elements are related to the generation of infected individuals in compartment $i$ by those in $j$. The $\Omega(t, \tau)$ matrix usually encompasses the susceptible compartments and parameters of the disease transmission. For the case in which $\Omega$ does not depend on $\tau$, as in ODE models, $\mathcal{F}_i(t) = \sum_j \Omega_{ij}(t)\,X_j(t)$, so that

$$\Omega_{ij}(t) = \frac{\partial}{\partial X_j}\mathcal{F}_i(t). \tag{2.9}$$

## 2.2. Obtaining the reproduction numbers

In order to estimate the reproduction number using the available incidence data, we need to link the equations of the model with the database. First, let us imagine that we are able to access a perfect dataset with the exact number of new infected at each calendar time, i.e. $\mathcal{F}(t)$. Of course, such perfect sets are hardly available from the health systems, but later in this section (§2.4), we will draw parallels between $\mathcal{F}(t)$ and the actually available data.

Since we established the $\mathcal{F}(t)$ form in (2.8), we want to play with it in order to get rid of the dependence on the compartments $x(t, \tau)$. Luckily, this can be easily done by substituting (2.7) in (2.8)

$$\mathcal{F}(t) = \int_0^\infty A(t, \tau)\mathcal{F}(t - \tau)\,\mathrm{d}\tau, \tag{2.10}$$

whereby $A(t, \tau)$ is expressed by the matrix product

$$A(t, \tau) = \Omega(t, \tau)\Gamma(t, \tau). \tag{2.11}$$

Analogously to [7], the functions $A_{ij}(t, \tau)$ represent the rate of new infections in $i$ due to previously infected $j$ individuals with an infection-age $\tau$, whereby $A_{ij}(t, \tau > t) \equiv 0$. Thus, we can account for the new cases in $i$ caused by cases that occurred previously in the other compartments. In fact, (2.10) is a general form for the widely known renewal equation [1,7], and can be interpreted as a sum of renewal equations. In order to separate each term of this sum, we define $\mathcal{J}_{ij}(t)$, i.e. the rate of new infections in the $i$ compartment due to previously infected $j$ individuals as

$$\mathcal{J}_{ij}(t) = \int_0^\infty A_{ij}(t, \tau)\mathcal{F}_j(t - \tau)\,\mathrm{d}\tau, \tag{2.12}$$

and emphasize that $\mathcal{F}_i(t) = \sum_j^n \mathcal{J}_{ij}(t)$. With these definitions, we can proceed to obtain the reproduction number. However, let us note that, when considering systems with multiple compartments as we do, the former reproduction number is replaced by the $n \times n$ next generation matrix $\mathcal{R}$ [15]. In this matrix, each element $\mathcal{R}_{ij}$ represents the expected number of new infections in $i$ generated by a newly infected individual at $j$. From (2.12), it becomes clear that we can get $\mathcal{R}_{ij}$ by integrating $A_{ij}(t, \tau)$ from zero to infinity with respect to $\tau$, as similarly done in [7], i.e.

$$\mathcal{R}_{ij}(t) = \int_0^\infty A_{ij}(t, \tau)\,\mathrm{d}\tau. \tag{2.13}$$

The basic reproduction number of the system is usually calculated from the spectral radius of this matrix at the disease-free fixed point. Actually, in terms of the basic reproduction number, our method is equivalent to the widely known van den Driessche next-generation method, [12]. Naturally, from

(2.13), we can also define

$$g_{ij}(t, \tau) = \frac{A_{ij}(t, \tau)}{\int_0^\infty A_{ij}(t, \tau)\,\mathrm{d}\tau},\tag{2.14}$$

such that $A_{ij}(t, \tau) = \mathcal{R}_{ij}(t)\,g_{ij}(t, \tau)$. The normalized elements $g_{ij}(t, \tau)$ are known as the generation interval distribution [1,7]. They are related to the flow of individuals between infected compartments and their recover process. We assume a generation interval distribution that might depend on $t$ and $\tau$, which is general enough to be applied to models with time-dependent parameters [16]. Therefore, using (2.13) and (2.14) in (2.12), we obtain

$$\mathcal{J}_{ij}(t) = \mathcal{R}_{ij}(t)\int_0^\infty g_{ij}(t, \tau)\,\mathcal{F}_j(t - \tau)\,\mathrm{d}\tau.\tag{2.15}$$

It is important to highlight that $\mathcal{R}_{ij}(t)$ is not necessarily the reproduction number that $j$ generates in $i$. Instead, the meaning of $\mathcal{R}_{ij}(t)$ is linked to the generation of new infected ones in $i$, $\mathcal{F}_i(t)$, due to infected individuals previously generated in $j$, $\mathcal{F}_j(t - \tau)$, regardless of the disease stage these $j$ individuals are at $t$. Next, to obtain the reproduction number a newly infected in $j$ is expected to generate in all other compartments, we just have to sum $\mathcal{R}_{ij}$ over $i$, which leads to

$$\overline{\mathcal{R}}_j(t) = \sum_i^n \mathcal{R}_{ij}(t).\tag{2.16}$$

From this point on, we adopt the notation in (2.16), where an over-line represents the collapse over the first index of matrix elements, in this case in the form of a sum over $i$. This leads us to analogously define $\overline{A}_j = \sum_i^n A_{ij}$, whereby its integral from zero to infinity with respect to $\tau$ corresponds to $\overline{\mathcal{R}}_j$, i.e.

$$\overline{\mathcal{J}}_j(t) = \overline{\mathcal{R}}_j(t)\int_0^\infty \overline{g}_j(t, \tau)\,\mathcal{F}_j(t - \tau)\,\mathrm{d}\tau,\tag{2.17}$$

where $\overline{\mathcal{J}}_j(t) = \sum_i^n \mathcal{J}_{ij}(t)$ and

$$\overline{g}_j(t, \tau) = \frac{\overline{A}_j(t, \tau)}{\int_0^\infty \overline{A}_j(t, \tau)\,\mathrm{d}\tau} = \frac{\sum_i^n \mathcal{R}_{ij}(t)g_{ij}(t, \tau)}{\sum_i^n \mathcal{R}_{ij}(t)}.\tag{2.18}$$

Therefore, $\overline{\mathcal{J}}_j(t)$ represents the rate of new infections generated by previous infections in $j$. Note that the generation interval distribution $\overline{g}_j(t, \tau)$ takes the form of an weighted average over the reproduction numbers, with weights given by $g_{ij}(t, \tau)$.

Summarizing, the general implementation of the proposed method consists of the following steps: identifying the terms $\mathcal{F}$ and $\mathcal{V}$ from a model; using them to find $\Omega$ and $\Gamma$; obtaining $A$ and integrating it to get $\mathcal{R}$ and $g$. Further in this work, we present applications of the method and estimations using actual data. Examples of the method applied to different types of models can be found on electronic supplementary material, S1.

## 2.3. The total reproduction number

Based on the pairwise reproduction numbers, we now define the total reproduction number $\mathcal{R}^T(t)$ for the whole system. The rate of new infections from a compartment, $\mathcal{F}_i(t)$, can be described as a fraction of the total rate from all compartments, $\mathcal{F}^T(t) = \sum_i^n \mathcal{F}_i(t)$ such that

$$\mathcal{F}_i(t) = \alpha_i(t)\mathcal{F}^T(t).\tag{2.19}$$

$\alpha_i(t)$ is the proportion of the total rate of new infections that $\mathcal{F}_i(t)$ represents, with the condition $\sum_i \alpha_i(t) = 1$. Thus, combining (2.10) with (2.19), we obtain

$$\mathcal{F}^T(t) = \mathcal{R}^T(t)\int_0^\infty g^T(t, \tau)\mathcal{F}^T(t - \tau)\,\mathrm{d}\tau\tag{2.20}$$

where

$$g^T(t, \tau) = \frac{\sum_i \alpha_i(t)\overline{\mathcal{R}}_i(t)\,\overline{g}_i(\tau)}{\sum_i \alpha_i(t)\overline{\mathcal{R}}_i(t)}.\tag{2.21}$$

Note that the total reproduction number of the system can be written as the scalar product between $\boldsymbol{\alpha}$ and $\overline{\mathcal{R}}$, i.e. $\mathcal{R}^T(t) = \boldsymbol{\alpha} \cdot \overline{\mathcal{R}}$. Thus, we can interpret it as a linear combination of the $\overline{\mathcal{R}}$, with the fractions $\boldsymbol{\alpha}$ acting as weights. Interestingly enough, it reveals how the system's behaviour is averaged over its heterogeneities. However, we point out that, as the given definition of $\mathcal{R}^T$ is very general, and it may not always have a meaningful interpretation. For instance, if we consider a system with two completely independent dynamics such that ($\mathcal{R}_{ij} = 0$ for $i \neq j$), it is still possible to evaluate $\mathcal{R}^T$, even if it has no meaning at all. Therefore, the $\alpha_i(t)$ functions play a key role in analysing whether the total reproduction number has a dynamical meaning.

We now focus our attention on a case where the elements $\alpha_i(t)$ appear quite naturally, by assuming that any $\Omega_{ij}(t, \tau)$ is given by a product of two functions, where one of them depends only on $i$ and the other only on $j$. This assumption occurs very often in disease transmission models, as is the case of both SIR and SEIR models. This general property can be expressed as

$$\Omega = \boldsymbol{\alpha} \otimes \overline{\Omega} = \left[ \alpha_i(t)\overline{\Omega}_j(t, \tau) \right], \tag{2.22}$$

where $\otimes$ represents a tensor product and $\overline{\Omega}_j = \sum_i \Omega_{ij}$. We note that $\overline{A}_j = \sum_k \overline{\Omega}_k \Gamma_{kj}$, so that $A = \boldsymbol{\alpha} \otimes \overline{A}$. In fact, the above equation also impacts equations (2.16), (2.17) and (2.18). The first and second ones can also be respectively, factorized as $\mathcal{R} = \boldsymbol{\alpha} \otimes \overline{\mathcal{R}}$ and $\mathcal{J} = \boldsymbol{\alpha} \otimes \overline{\mathcal{J}}$, while the result for the last one can be simplified to $g_{ij}(t, \tau) = \overline{g}_j(t, \tau)$. Furthermore, because the next generation matrix is obtained from a tensor product of vectors, the spectral radius of $\mathcal{R}$ corresponds to the scalar product of $\overline{\mathcal{R}}$ and $\boldsymbol{\alpha}$, that is $\mathcal{R}^T$ [12]. Thus, in these systems the total reproduction number evaluated at the disease-free equilibrium point, $t = 0$, corresponds to the basic reproduction number, $\mathcal{R}^T(0) = \mathcal{R}_0$.

## 2.4. Estimations with real data

So far, we have developed a general framework to estimate the reproduction numbers from the rate of new infections $\mathcal{F}$. Now, let us establish a connection between $\mathcal{F}$ and the available data, starting by defining the elements of the vector $\mathcal{B}$ as

$$\mathcal{B}_i(t) = \int_t^{t+\Delta t} \mathcal{F}_i(t') \, dt'. \tag{2.23}$$

$\mathcal{B}_i(t)$ represents the number of new infections in $i$ between $t$ and $t + \Delta t$. The interval $\Delta t$ should reflect the notification frequency of the data, i.e. days, weeks and so on. We defined the $\mathcal{B}_i(t)$ function because the real-world incidence data is usually a time series with the collection of all reported cases in a $\Delta t$ period of time. However, there are important differences between a reported case and a new infection $\mathcal{B}_i(t)$. For a new infection to become a reported case, the individual needs to test positively for the disease, in a process subject to errors and delays. First, the individual needs to present enough clinical symptoms, which usually take some time to develop. Next, the result of the tested individual is not always immediately registered into the database, leaving a time gap between the test and the registration of the case. To deal with such deficiencies, especially in conditions where such delays are not negligible, specific correcting techniques directly applied to the reported data have been developed, like nowcasting [17] and back projection [18,19], which provide better estimates of $\mathcal{B}_i(t)$. Nevertheless, we emphasize that difficulties in accessing good quality data are not directly related to the method introduced herein.

Thus, after wishfully improving the quality of the estimates $\mathcal{B}_i$, we proceed by considering the discrete form of (2.15) [20]

$$\mathcal{T}_{ij}(t) = \mathcal{R}_{ij}(t) \sum_{\tau=0}^{t} g_{ij}(t, \tau) \mathcal{B}_j(t - \tau) \Delta t, \tag{2.24}$$

whereby

$$\mathcal{T}_{ij}(t) = \int_t^{t+\Delta t} \mathcal{J}_{ij}(t') \, dt'. \tag{2.25}$$

Of course, $\mathcal{B}_i(t) = \sum_j^n \mathcal{T}_{ij}$. Equation (2.24) is a generalization of the discrete version of the renewal equation. In fact, by using (2.20), we are able to recover the form of a well-known result in the

literature [20]

$$\mathcal{R}^T(t) = \frac{\mathcal{B}^T(t)}{\sum_{\tau=0}^{t} g^T(t, \tau)\mathcal{B}^T(t - \tau)\Delta t}, \tag{2.26}$$

where $\mathcal{B}^T(t) = \sum_{i}^{n} \mathcal{B}_i(t)$.

# 3. Explicit expression of $\mathcal{R}(t)$ for two meta-population models

In this section, we apply the methodology developed in this study to obtain the reproduction numbers and correspondent generation interval distributions for two meta-population models. Details of the necessary calculations and of the used models are available in electronic supplementary material, S1 and S2.

## 3.1. SIR-type meta-population model

In this subsection, we consider a meta-population model that takes into account groups of spatially separated 'island' populations with interactions. Such models are widely used in the context of spatiotemporal disease spread [21–27]. A system like this can be treated as a network in which the nodes represent the meta-populations while the weighted edges between them represent the intensity of their interaction. Here, the meta-populations' interactions result from the commuter movement of individuals between their residence, work and study places. This type of movement is obligatory cyclical, predictable and recurring regularly, most of time on a daily basis. Thus, the population of each node does not change with time, since the individuals always comes back to their original residence place.

In this model, we assumed that each meta-population $i$, with $N_i$ individuals, has its own transmission rate $\beta_i(t)$. The movement of individuals between meta-populations is described by the density of flow from $i$'s population to $j$, $\Phi_{ij}(t)$, i.e. the number of $i$ resident individuals commuting from $i$ to $j$ divided by $N_i$. All meta-populations are assumed to have the same recovery dynamics, i.e. the same recovery rate $\gamma$. In electronic supplementary material, S2, we present details of this model inspired by [28]. However, it is important to call attention to a significant aspect of the current approach, the main goal of which is the daily evaluation of $R(t)$, not predicting the evolution of the epidemic. To this purpose, it is necessary to account for the daily fluctuations observed in the used data as they interfere in the actual value of $R(t)$. Therefore, we let the parameters $\beta_i(t)$ and $\phi_{ij}(t)$ be time dependent, the daily values of which being evaluated based on the application of the methodology. The same argument is valid for the model SEIIR we discuss in the next subsection.

The reproduction numbers $R_{ij}(t)$ and generation interval distributions $g_{ij}(t)$ for this model are expressed by

$$R_{ij}(t) = S_i(t)\frac{\lambda_{ij}(t)}{\gamma}, \quad g_{ij}(\tau) = g(\tau) = \gamma e^{-\gamma\tau}. \tag{3.1}$$

The details of the derivations of these expressions are presented in electronic supplementary material, S1. Here, $\lambda_{ij}$ is related to the transmission of the disease from a meta-population $j$ to another meta-population $i$ and is derived based on simple assumptions about the commuter movement of individuals in the network (see electronic supplementary material, S2). The $g_{ij}(\tau) = g(\tau)$ relation appears naturally from the assumption that all meta-populations have the same recovery dynamics. Noteworthy, if we isolate the meta-populations in the network, $\Phi_{ij}(t) = 0$, $\forall i$ and $j$, all reproduction numbers and the generation interval distributions become identical to that of the classical SIR model [7].

## 3.2. A meta-population model for COVID-19 (SEIIR)

Here, we focus on a meta-population model for a specific disease, the SARS-CoV-2 coronavirus. In this case, the transmission can be facilitated by the existence of individuals whose symptoms are very weak or even non-existent [28]. In order to have a consistent description of this aspect, the model considers the existence of two classes of infected individuals, the symptomatic and the asymptomatic/undetected ones, as considered in a more general model for the same disease [29]. Therefore, it also accounts for infected individuals not needing to be hospitalized. Usually, they are not included in any officially registered data, thus becoming undetectable. For the sake of simplicity, we will refer to such

individuals only as asymptomatic. In electronic supplementary material, S2, we indicate how to derive this model based on the meta-population SIR-type approach described in the previous section. In electronic supplementary material, S1, the expressions for the reproduction number and generation interval distribution are derived in detail. There it is shown that we only need the elements of $\mathcal{R}$ with $i, j \leq n$ to describe the dynamics. Thus, in this main framework, whenever we refer to $\mathcal{R}$ or $g$ we are alluding to their $i, j \leq n$ elements, which read

$$R_{ij}(t) = S_i(t)\lambda_{ij}(t)\left[\frac{p}{\gamma_s} + \frac{\delta(1-p)}{\gamma_a}\right] \quad \text{and} \quad g_{ij}(\tau) = g(\tau) = \frac{(p/\gamma_s)g^s(\tau) + (\delta(1-p)/\gamma_a)g^a(\tau)}{(p/\gamma_s) + \delta(1-p)/\gamma_a}. \tag{3.2}$$

Note that the expressions for $\lambda_{ij}$ are the same obtained for the SIR-type model. Other model parameters have the following meanings: $\delta$—a factor that reduces or enhances the asymptomatic infectivity; $p$—the proportion of individuals that becomes symptomatic when infected; $\gamma_a$ and $\gamma_s$—the recovery rates of the asymptomatic and symptomatic individuals, respectively. $g^a(\tau)$ and $g^s(\tau)$ are expressed in terms of exponential functions as

$$g^a(\tau) = \frac{\kappa\gamma_a}{\gamma_a - \kappa}(e^{-\kappa\tau} - e^{-\gamma_a\tau}) \quad \text{and} \quad g^s(\tau) = \frac{\kappa\gamma_s}{\gamma_s - \kappa}(e^{-\kappa\tau} - e^{-\gamma_s\tau}). \tag{3.3}$$

Similarly to what was observed before, the reproduction numbers and generation interval distributions collapse to the corresponding expressions obtained [29] when all meta-populations in the network are isolated by setting $\Phi_{ij}(t) = 0$, $\forall i$ and $j$.

# 4. Applications for the meta-population models using actual data

In this section, we present numerical results for the two models discussed in §3 using actual data on the first six months of the COVID-19 pandemic in a set of Brazilian cities forming the metropolitan area of Rio de Janeiro. The dataset comprises the following records: reported cases in each municipality, daily commuter movement due to work between municipalities, and daily mobility tends towards workplaces. In electronic supplementary material, S3, we derive the expressions and parameters needed to estimate a daily time series of the reproduction numbers for each model.

## 4.1. Database

We used daily notifications of new cases due to COVID-19 in Brazil obtained from two public websites: https://covid.saude.gov.br/ and https://brasil.io/datasets/. The original data were provided by the Brazil Health Ministry. Data for the inter-municipality commuter movement of workers and students were obtained from a study on population arrangements and urban concentrations in Brazil conducted by IBGE (Brasilian Institute of Geography and Statistics) in 2015, which can be found in [30]. In addition, we obtained daily mobility data for each Brazilian state from a public report by Google, accessed at https://google.com/covid19/mobility/.

In order to estimate the number of new infections out of the reported data, we use a back-projection method, introduced in [19]. The method is executed using a diagnosis distribution, which gives the probability of a given delay between infection and testing [31]. This distribution is the convolution of two other known ones: the incubation distribution, which gives the probability for a delay between infection and symptoms; testing distribution, which gives the probability for a delay between symptoms and testing. The incubation distribution for the SARS-CoV-2 has been estimated in [32], and the testing distribution is assumed to be log-normal, as in [31], with mean and standard deviation of that time delay being $10.1 \pm 17.1$ days, estimated for the southeast region of Brazil where the state of Rio de Janeiro is located, during the first six months of COVID-19 pandemics as it was shown in [33]. The implementation of the back-projection method is done using the programming language R with the function backprojNP of the package Surveillance, available at https://rdrr.io/github/jimhester/surveillance/. The maximum delay assumed for the distribution was of 30 days and we performed a 10-day moving average in order to attenuate noise and better express the data trend.

To take into account the social distancing restrictions, we considered only the commuter movement data related to work. Indeed, due to adopting mitigation measures to control the spread of COVID-19, the inter-municipality flows for education purposes were significantly reduced. In addition, the movement of workers towards their working places was estimated by using the mobility data obtained from the community mobility report provided by Google. This database compares, for each

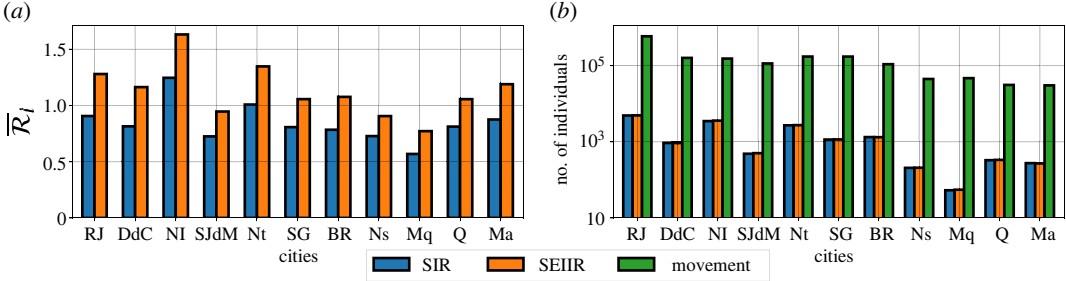

**Figure 1.** Comparison between SIR and SEIIR outputs. SIR results in blue and SEIIR ones in orange. The bar graph (a) compares the SIR and SEIIR $\overline{\mathcal{R}}_i$ time averages for all 11 municipalities. In (b), for each municipality, estimations for the total number of exported infections that are reported for both models are displayed with the total commuter movement in that city. The names of the municipalities are abbreviated using acronyms: Rio de Janeiro (RJ), Duque de Caxias (DdC), Nova Iguaçu (NI), São João de Meriti (SJdM), Niterói (Nt), São Gonçalo (SG), Belford Roxo (BR), Nilópolis (Ns), Mesquita (Mq), Queimados (Q), Magé (Ma).

Brazilian state, the daily mobility to workplaces with the past trends. In this way we obtained the reduction in work commuting. A moving average was performed in the mobility time series, and the inter-municipality workflow was reduced according to the percentage indicated on the data. All these factors were included in the evaluation of $\Phi_{ij}(t)$. The parameters used to feed the model were obtained in [34], and can be found in electronic supplementary material, S3.

The used model gathers data for 11 cities, starting with Rio de Janeiro (RJ), Brazil and, in addition, the following 10 smaller cities of its metropolitan area with the largest commuter flow with it: Duque de Caxias (DdC), Nova Iguaçu (NI), São João de Meriti (SJdM), Niterói (Nt), São Gonçalo (SG), Belford Roxo (BR), Nilópolis (Ns), Mesquita (Mq), Queimados (Q), Magé (Ma). Additional information about the municipalities is presented in electronic supplementary material, S4.

## 4.2. Analyses of the results

In our first results, shown in figure 1, we present a comparison between the SIR and SEIIR outputs. Using the daily time series of the reproduction numbers (see electronic supplementary material, S3), we obtain the series of $\overline{\mathcal{R}}(t)$ and $\mathcal{T}_{ij}(t)$ whose elements are given by equations (2.16) and (2.24), respectively. We observe that values of elements of $\overline{\mathcal{R}}$ using the SEIIR model are, on the average, 33% higher then their counterparts for the SIR model. On the other hand, the estimations for the total numbers of exported infections generated by a city $j$, $\sum_t \sum_i^n \mathcal{T}_{ij}(t)$ for $i \neq j$, are very similar for both SIR and SEIIR models. Also, it seems that the total commuter movement, which is the sum of all the inflows and outflows occurring in a municipality, is not the only main factor that determines the number of exported infections of a municipality. This nonlinear effect can be observed when comparing SJdM and Nt or DdC and NI (figure 1b). Those municipalities have a similar amount of total flow but very different results for the exported infections. Interestingly, even not having the highest $\overline{\mathcal{R}}_j$, the biggest city, RJ, presents the largest amount of exported infections, which also highlights the nonlinear dynamics of the phenomenon.

From now on, we report only results for the SEIIR model since it provides a more realistic description of the dynamics of COVID-19. Using $\mathcal{T}_{ij}(t)$, we can access the contribution of each municipality to the outbreaks happening in the state. Thus, by dividing $\mathcal{T}_{ij}(t)$ by $\mathcal{B}_i(t)$ in every time step, we obtain a time series for the fraction of the total infections in $i$ generated by $j$. This way we evaluate the time evolution of the mean value of $\mathcal{T}_{ij}/\mathcal{B}_i$, as displayed in figure 2, where the numerator and denominator are, respectively, given by (2.23) and (2.25). It is observed a high autochthonous behaviour on the disease transmission, indicating that the highest influence on disease transmission of a city is on itself, i.e. most of the infections generated in a municipality are caused by its own individuals. However, we also identify cities where the infections generated by other municipalities on it are very important. The $\mathcal{R}(t)$ matrix also corroborates the presence of an important autochthonous behaviour, as its diagonal elements correspond to the highest values of the reproduction numbers. We also observe a large number of very small off-diagonal elements in the matrix.

In figure 3, we display again the most relevant findings in figure 2, by selecting only non-autochthonous influences above 5%. Main features include RJ as the most important agent causing disease transmission to other municipalities in the network. However, cities like NI also present

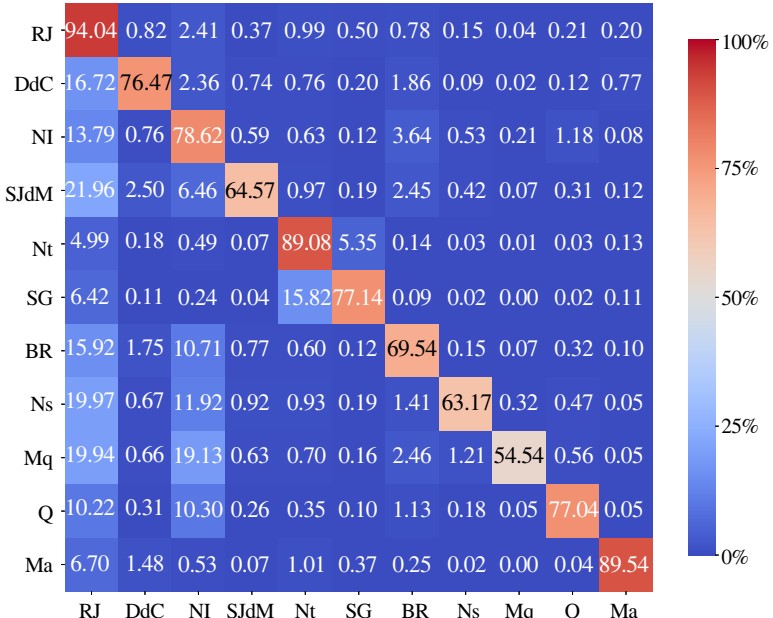

**Figure 2.** Influence on number of infections caused by a municipality on another. The heat map captures the time-averaged influence on the number of infections caused by a municipality on another as a fraction of the total number of daily infections, $\mathcal{T}_{ij}/\mathcal{B}_i$.

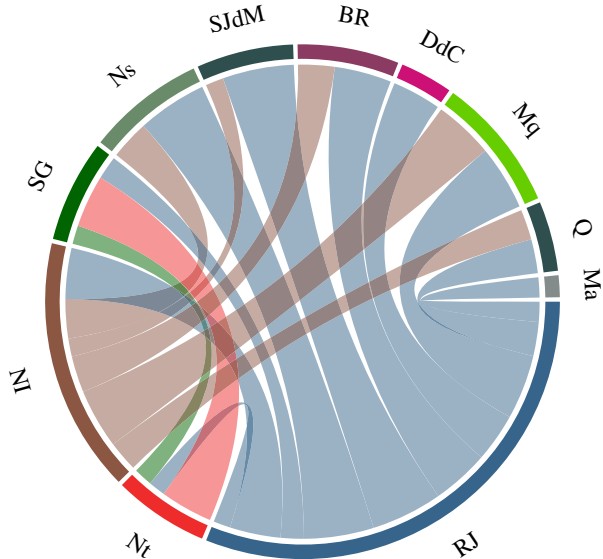

**Figure 3.** Visualization of mutual municipality influences. Here, we provide a visualization of the most relevant results from figure 2. Only non-autochthonous influences above 5% were considered. The thickness of the lines connecting municipalities is proportional to the number of infections that one generates on the other. The colour of each line represents the municipality responsible for generating the infections.

themselves as a relevant spreader of the pathogen. As shown in figure 1, NI is the city with largest contribution in infection exportation after RJ. In figure 3, we identify that cities like SJdM, BR, Ns, Mq and Q are the main receptors of these infections. Nt, even having a NI-like number of exported infections, did not present a large influence on many cities. On the other hand, Nt generates a significant amount of infections in SG, highlighting the importance of the connection between these two municipalities.

In figure 4, we illustrate the time series reconstruction resulting from the SEIIR model, focusing on two different municipalities. In the first one, we focus on the infections in SG and compare the total amount, $\mathcal{B}_i(t)$, with the model prediction of the number of daily infections in SG generated by RJ and

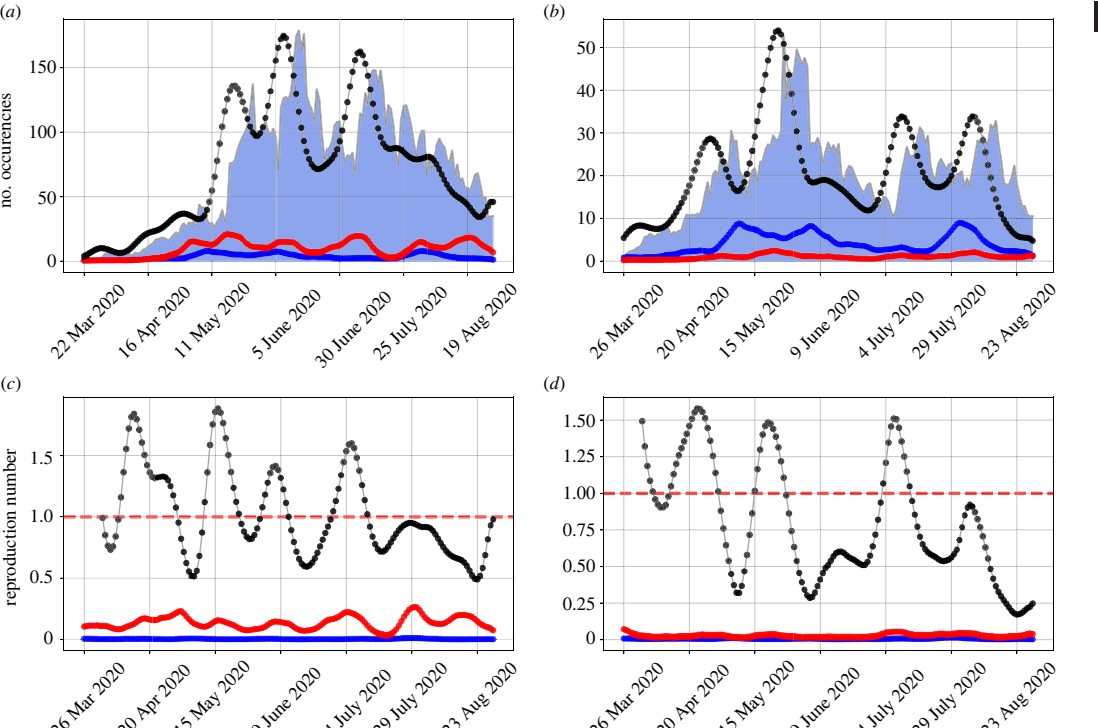

**Figure 4.** Reconstruction of the time series and reproduction numbers for SG (*a,c*) and SJdM (*b,d*). Black dots represent, in (*a,b*), the daily new infections and, in (*c,d*), the sum of all the reproduction numbers related to new infection on the municipality, i.e. $\sum_j \mathcal{R}_{ij}$. Blue dots indicate, in (*a,b*), the amount of new infections generated in the respective cities due to commute flow with RJ and, in (*c,d*), the reproduction number related to the new infections in the municipality due to RJ. Red dots correspond, in (*a,b*), to the number of new infections due to Nt in SG (*a*), and due to NI in SJdM (*b*), and, in (*c,d*), to the reproduction number related to the new infections in the municipality due to Nt in SG (*c*), and due to NI in SJdM (*d*). In (*a, b*), the grey lines contouring the blue shaded areas indicate the number of new infections of that given municipality.

Nt. We observe that Nt exerts a larger influence over SG than RJ, contributing with a higher number of generated infections at all times. In addition, we present the sum of all the reproduction numbers related to new infection on the municipality, i.e. $\sum_j \mathcal{R}_{ij}$, and compare it with the contributions related to RJ and SG. We observe that the reproduction number related to SG is higher than the one related to RJ for all values of time. The second scenario is related to the total number of infections in SJdM and the model contributions due to RJ and NI. In this case, RJ contributes with the largest number of infections generated in that city, besides the city itself. The reproduction number related to the new infections in SJdM due to RJ is lower than the one of NI. Thus, even with a lower reproduction number, RJ is able to generate the highest amount of infections in SJdM due to the huge amount of infected individuals in it.

## 5. Discussion

This work provides a theoretical tool for the study and investigation of the infectious disease spread within the scope of heterogeneous compartmental models. By starting from an arbitrary compartmental model, we show how to construct a very general method for obtaining analytical expressions for the reproduction numbers and generation interval distributions. The methodology is very general and has several applications in the scope of both modelling and data analysis. We show how to combine the theoretical structure with incidence data, establishing the path to estimate the reproduction numbers. These results can become the basis for several possible data analyses. The method is robust and reproduces known results in the literature, as shown in electronic supplementary material, S1 for the SIR, SEIR and SEIIR models. It opens room for analyses of more sophisticated models, which aim at a better understanding and control of infectious disease processes mainly by allowing to measure the effective reproduction number between the sub-compartments. Note that once the method allows the infection rate to vary with time, the results capture the possible

changes of the transmission rate. In [29,34], it is shown that a less general version of this method is able to indicate the changes of the transmission rate due to the mitigation policies applied on the early stages of the SARS-CoV-2 pandemic. Due to this possible variation of the contamination process, assuming that $R(t) = R_0 * S(t)$ in all more complex situations does not lead to correct estimates.

In the second part of the work, we focused on providing an application for the methodology. Using SIR and SEIIR meta-population models, we obtained explicit expressions for their reproduction numbers and generation interval distributions. The $\mathcal{R}(t)$ for both models differ only by multiplying factors, $1/\gamma$ and $(p/\gamma_s) + (\delta(1-p)/\gamma_a)$, as in [29,34]. Combining the theoretical results with the reported data we evaluated, through SEIIR meta-population model, the role of each municipality on the COVID-19 spread on 11 cities at the metropolitan region of Rio de Janeiro, Brazil. Cities like Nova Iguaçu, Niterói and São Gonçalo pop out as important agents on the spread of the pathogen throughout the region. RJ itself plays the role of the main spreading hub, given its high disease incidence rate and its central role in the commuting displacement of individuals, as also observed in [27,34,35]. However, cities like Niterói and São Gonçalo, whose mutual interaction is larger than that they have with RJ, cannot be neglected. This highlights the importance of flow control between municipalities as an important strategy in the diffusion of the pathogen, especially in heterogeneous systems as previously addressed in network models [27].

In this work, we presented an analysis based on officially reported data, taking into account the diagnosis distribution, based on back-projection method, to estimate the new infections out of the reported data. Our results are not limited to reporting the reproduction number, emphasizing that its value is not a dead-end result. Going beyond that, we explicitly demonstrated how our method leads to deeper analyses, such as the reconstruction of the time series and the evaluation of the number of exported infections.

Of course, our results for the Rio de Janeiro metropolitan area can be further extended, by including other features left out in this first approach, in which attesting the reliability of the developed method was the main focus. For instance, more sophisticated commuter flow models available in the literature can provide a more precise description of pendular behaviour. Other possibilities consist in including the influence of international and interstate passenger traffic, and further relevant heterogeneous features related to age, social heterogeneities and segregation. Note that the latter would also be included in our general theoretical formalism. Since the frequency of testing changes over the course of an epidemic, assuming a fixed testing distribution for the back-projection calculations may bring limitations to our estimate of the infection curve [36]. The basic compartment model used in our analyses can also be extended to include self-quarantine of the infected symptomatic individuals, which surely plays an important role in disease propagation. The limitations in the quality of data certainly affect the outputs produced by any used model, ours included, but such source of problem can be overcome in future studies by accounting for the uncertainties regarding the data and parameters.

In spite of the limitations of the discussed results, they do provide new features and clarifications for the analysed system. They also indicate that the robust theoretical framework developed herein may contribute to further advances in mathematical modelling, given its broad applicability to a large class of infectious disease spreading models.

Ethics. Since all data handled in this study are publicly available, approval by an ethics committee is not required, according to Resolutions 466/2012 and 510/2016 (article 1, sections III and V) from the National Health Council (CNS), Brazil.

Data accessibility. In this work, we use daily notifications of new cases due to COVID-19 in Brazil, obtained from public websites: https://covid.saude.gov.br/ and https://brasil.io/datasets/, which provide data from the Health Ministry. We obtained intermunicipal commuter movement of workers and students data from a study on population arrangements and urban concentrations in Brazil conducted by IBGE (Brasilian Institute of Geography and Statistics) in 2015 that can be found at: https://www.ibge.gov.br/geociencias/organizacao-do-territorio/divisao-regional/15782-arranjospopulacionais-e-concentracoes-urbanasdo-brasil.html?=&t=o-que-e. In addition, we obtained daily mobility data for each Brazilian state from a public report by Google, accessed at: https://google.com/covid19/mobility/. Data and relevant code for this research work are stored in GitHub: https://github.com/danielcpj/Rt-heterogeneous-models and have been archived within the Zenodo repository: https://doi.org/10.5281/zenodo.5822669 [37].

Supplementary material is available online [38].

Authors' contributions. D.C.P.J.: conceptualization, data curation, formal analysis, investigation, methodology, validation, visualization, writing—original draft, writing—review and editing; J.F.O.: conceptualization, investigation, methodology, writing—review and editing; J.G.V.M.: conceptualization, methodology, writing—review and editing; R.F.S.A.: conceptualization, formal analysis, investigation, methodology, supervision, writing—review and editing; S.T.R.P.: conceptualization, formal analysis, investigation, methodology, supervision, writing—review and editing.

All authors gave final approval for publication and agreed to be held accountable for the work performed therein. Conflict of interest declaration. We declare we have no competing interests.

Funding. D.C.P.J. was funded by a Scientific Initiation scholarship from CNPq (process no. 117568/2019-8) and a Master's degree scholarship from FAPESP (process no. 2020/15643-8). J.F.O. was supported by the Fiocruz Program of Promotion of Innovation—innovative ideas and products—COVID-19, orders and strategies—Inova Fiocruz (Processo VPPIS-005-FIO-20-2-40), and the Center of Data and Knowledge Integration for Health (CIDACS) through the Zika Platform—a long-term surveillance platform for Zika virus and microcephaly (Unified Health System (SUS), Brazilian Ministry of Health). S.T.R.P. was supported by an International Cooperation grant (process no. INT0002/2016) from Bahia Research Foundation (FAPESB). R.F.S.A. was supported by Brazilian agency CNPq through grants nos. 422561/2018-5 and 304257/2019-2. S.T.R.P. and R.F.S.A. were supported by the National Institute of Science and Technology—Complex Systems from CNPq, Brazil. J.G.V.M. acknowledges the support of the National Council of Technological and Scientific Development, CNPq, Brazil (grant no. 307828/2018-2).

Acknowledgements. The authors acknowledge the discussions and suggestions from members of the CoVida Network (http://www.redecovida.org).

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
