## [Peer Review File · Royal Society Open Science]

Review History

RSOS-210700.R0 (Original submission)

Review form: Reviewer 1

Is the manuscript scientifically sound in its present form?

Yes

Are the interpretations and conclusions justified by the results?

Yes

Is the language acceptable?

No

Do you have any ethical concerns with this paper?

No

Have you any concerns about statistical analyses in this paper?

No

Recommendation?

Major revision is needed (please make suggestions in comments)

Comments to the Author(s)

In this manuscript, the authors present a general approach to deriving the generation interval distribution and the time-varying reproductive number within an ODE model that includes host heterogeneity (e.g. heterogeneity in age, space, risk group). This builds on previous work describing how to calculate the basic (but not the time-varying) reproductive number in models of simple or heterogeneous populations. It also builds on previous work describing how to derive the generation interval within an SEIR-type model (without heterogeneity) (<https://doi.org/10.1137/18M1186411>).

From what I can assess, the work seems sound, and useful. Differential equations are not my area of expertise, and I was not able to assess every detail of the derivations presented here. However, throughout the derivation I can follow the general logic, and can draw mathematical parallels between the equations/approach presented here, and methods I've used before when working with simpler SEIR-type, or empirical models for estimating $R(t)$. The general logic and mathematics of the derivations shown in section 2 seem reasonable, and they simplify in ways consistent with previously published work.

1. The main weakness of the submission is that, at several key points in the text, the writing is difficult to understand. I think it's unfair to researchers in non-English speaking countries that publications are almost always written in English. But I encourage the authors to seek editorial help from a native English speaker if possible. Overall, the scientific content seems good, but the impact of this paper will be greater if the ideas are easier for readers to understand (more specifics are discussed below).

2. The significance of the work, and its relationship to existing methods, is not explained very clearly in the abstract/introduction.

It would be helpful to draw a clearer distinction between empirical approaches to estimating $R(t)$ and the generation interval, and inference of $R(t)$ within compartment models, where the generation interval must be derived from the equations of the model. Empirical methods are not really the focus of this manuscript – these methods are already well-established and widely used (pre-2020 R_t methods reviewed in - <https://doi.org/10.1371/journal.pcbi.1008409>, generation interval estimation - <https://doi.org/10.1073/pnas.2011548118>, newer R_t methods - <https://doi.org/10.1101/2020.09.14.20194589>, see also the recently developed R packages EpiNow2, Epifilter, Epidemia, and others).

- From the abstract -- “However, there are systems, especially highly heterogeneous ones, in which there is a lack of data and an adequate methodology to obtain $g(\tau)$.” As currently written, this claim seems overly broad, as it's not clear if the manuscript is trying to argue that:
 - a. There are no adequate methods to estimate $g(\tau)$ empirically from surveillance data, in which heterogeneities between hosts exist inevitably. (this statement is not true, see -- <https://doi.org/10.1073/pnas.2011548118>; <https://doi.org/10.1016/j.epidem.2018.12.002>), or
 - b. Methods to derive the generation interval distribution from a compartment model that includes host heterogeneity have not been established (to my knowledge, this is true, and I think it's what the authors are trying to say).

- It would be helpful to clarify what the authors mean by “highly heterogeneous systems”. For example, making it clear that the authors are referring to models that include heterogeneities such as age, geographic or risk structure in the host population would be helpful.

- Clarify that this method can produce separate $R_{ij}(t)$ estimates for transmission within and between model sub-compartments. (This is innovative!)

3. This may be obvious to the authors, but why isn't it correct/sufficient to estimate R_0 using the next generation matrix method, and then obtain $R(t)$ using the equation: $R(t) = R_0 * S(t)$? While not necessary for publication, it would be quite helpful for readers and users of these methods to see how badly this easier approach performs relative to the exact solution presented here.

4. Sometimes the notation is not explained clearly, or used inconsistently, which makes it difficult to follow the mathematical arguments. E.g. from page 4 -

- Is it true (25) that x can really be interpreted as the "infection age distribution?" In the previous sentence, the authors state that x is a vector whose entries represent individuals infected at the same time (i.e. of the same infection age), but from different population sub compartments.

- line 32 - the meaning of the V and F matrices could be explained more clearly. Readers already familiar with the next generation method should be able to follow, but re-wording would be helpful!

- Maybe this is a mathematical convention that I'm not familiar with, but at various points I found it confusing that the x notation switched between capital and lowercase letters. It seems like X is used to indicate totals across all τ values, but an explanation could be helpful.

5. It's interesting that $g_{ij}(\tau) = g(\tau)$ in all the examples presented in the supplement and main text. Can the authors make any general statements about when this is or is not true?

6. A point worth discussing is that empirically, the generation interval has been shown to change over time (DOI: 10.1126/science.abc9004), but these changes are not straightforward to include in compartment models, and not accounted for here.

7. Could also be worth discussing other methods used to study geographic spread of influenza and COVID-19 epidemics (e.g. gravity models or network models).

Minor - There are a few editing mistakes (e.g. repeated words, and one sentence in Portuguese) in the supplementary methods.

Review form: Reviewer 2

Is the manuscript scientifically sound in its present form?

No

Are the interpretations and conclusions justified by the results?

No

Is the language acceptable?

Yes

Do you have any ethical concerns with this paper?

No

Have you any concerns about statistical analyses in this paper?

No

Recommendation?

Reject

Comments to the Author(s)

See attached report (Appendix A).

Decision letter (RSOS-210700.R0)

Dear Mr Jorge

The Editors assigned to your paper RSOS-210700 "Estimating the effective reproduction number for heterogeneous models using incidence data" have made a decision based on their reading of the paper and any comments received from reviewers.

Regrettably, in view of the reports received, the manuscript has been rejected in its current form. However, a new manuscript may be submitted which takes into consideration these comments.

We invite you to respond to the comments supplied below and prepare a resubmission of your manuscript. Below the referees' and Editors' comments (where applicable) we provide additional requirements. We provide guidance below to help you prepare your revision.

Please note that resubmitting your manuscript does not guarantee eventual acceptance, and we do not generally allow multiple rounds of revision and resubmission, so we urge you to make every effort to fully address all of the comments at this stage. If deemed necessary by the Editors, your manuscript will be sent back to one or more of the original reviewers for assessment. If the original reviewers are not available, we may invite new reviewers.

Please resubmit your revised manuscript and required files (see below) no later than 05-Jan-2022. Note: the ScholarOne system will 'lock' if resubmission is attempted on or after this deadline. If you do not think you will be able to meet this deadline, please contact the editorial office immediately.

Please note article processing charges apply to papers accepted for publication in Royal Society Open Science (<https://royalsocietypublishing.org/rsos/charges>). Charges will also apply to papers transferred to the journal from other Royal Society Publishing journals, as well as papers submitted as part of our collaboration with the Royal Society of Chemistry (<https://royalsocietypublishing.org/rsos/chemistry>). Fee waivers are available but must be requested when you submit your manuscript (<https://royalsocietypublishing.org/rsos/waivers>).

Thank you for submitting your manuscript to Royal Society Open Science and we look forward to receiving your resubmission. If you have any questions at all, please do not hesitate to get in touch.

on behalf of Professor Tim Rogers (Associate Editor) and Mark Chaplain (Subject Editor)
openscience@royalsociety.org

Associate Editor Comments to Author (Professor Tim Rogers):

I recommend to follow the advice of referee 2 and redo the work with a more careful approach to data integration. The result will be a substantially new article which could be resubmitted here.

Reviewer comments to Author:

Reviewer: 1

Comments to the Author(s)

In this manuscript, the authors present a general approach to deriving the generation interval distribution and the time-varying reproductive number within an ODE model that includes host heterogeneity (e.g. heterogeneity in age, space, risk group). This builds on previous work describing how to calculate the basic (but not the time-varying) reproductive number in models of simple or heterogeneous populations. It also builds on previous work describing how to derive the generation interval within an SEIR-type model (without heterogeneity) (<https://doi.org/10.1137/18M1186411>).

From what I can assess, the work seems sound, and useful. Differential equations are not my area of expertise, and I was not able to assess every detail of the derivations presented here. However, throughout the derivation I can follow the general logic, and can draw mathematical parallels between the equations/approach presented here, and methods I've used before when working with simpler SEIR-type, or empirical models for estimating $R(t)$. The general logic and mathematics of the derivations shown in section 2 seem reasonable, and they simplify in ways consistent with previously published work.

1. The main weakness of the submission is that, at several key points in the text, the writing is difficult to understand. I think it's unfair to researchers in non-English speaking countries that publications are almost always written in English. But I encourage the authors to seek editorial help from a native English speaker if possible. Overall, the scientific content seems good, but the impact of this paper will be greater if the ideas are easier for readers to understand (more specifics are discussed below).

2. The significance of the work, and its relationship to existing methods, is not explained very clearly in the abstract/introduction.

It would be helpful to draw a clearer distinction between empirical approaches to estimating $R(t)$ and the generation interval, and inference of $R(t)$ within compartment models, where the generation interval must be derived from the equations of the model. Empirical methods are not really the focus of this manuscript - these methods are already well-established and widely used (pre-2020 R_t methods reviewed in - <https://doi.org/10.1371/journal.pcbi.1008409>, generation interval estimation - <https://doi.org/10.1073/pnas.2011548118>, newer R_t methods - <https://doi.org/10.1101/2020.09.14.20194589>, see also the recently developed R packages EpiNow2, Epifilter, Epidemia, and others).

- From the abstract -- "However, there are systems, especially highly heterogeneous ones, in which there is a lack of data and an adequate methodology to obtain $g(\tau)$." As currently written, this claim seems overly broad, as it's not clear if the manuscript is trying to argue that:
 - a. There are no adequate methods to estimate $g(\tau)$ empirically from surveillance data, in which heterogeneities between hosts exist inevitably. (this statement is not true, see -- <https://doi.org/10.1073/pnas.2011548118>; <https://doi.org/10.1016/j.epidem.2018.12.002>), or
 - b. Methods to derive the generation interval distribution from a compartment model that includes host heterogeneity have not been established (to my knowledge, this is true, and I think it's what the authors are trying to say).
- It would be helpful to clarify what the authors mean by "highly heterogeneous systems". For example, making it clear that the authors are referring to models that include heterogeneities such as age, geographic or risk structure in the host population would be helpful.
- Clarify that this method can produce separate $R_{ij}(t)$ estimates for transmission within and between model sub-compartments. (This is innovative!)

3. This may be obvious to the authors, but why isn't it correct/sufficient to estimate R_0 using the next generation matrix method, and then obtain $R(t)$ using the equation: $R(t) = R_0 * S(t)$? While not necessary for publication, it would be quite helpful for readers and users of these methods to see how badly this easier approach performs relative to the exact solution presented here.

4. Sometimes the notation is not explained clearly, or used inconsistently, which makes it difficult to follow the mathematical arguments. E.g. from page 4 -

- Is it true (25) that x can really be interpreted as the "infection age distribution?" In the previous sentence, the authors state that x is a vector whose entries represent individuals infected at the same time (i.e. of the same infection age), but from different population sub compartments.
- line 32 - the meaning of the V and F matrices could be explained more clearly. Readers already familiar with the next generation method should be able to follow, but re-wording would be helpful!
- Maybe this is a mathematical convention that I'm not familiar with, but at various points I found it confusing that the x notation switched between capital and lowercase letters. It seems like X is used to indicate totals across all τ values, but an explanation could be helpful.

5. It's interesting that $g_{ij}(\tau) = g(\tau)$ in all the examples presented in the supplement and main text. Can the authors make any general statements about when this is or is not true?

6. A point worth discussing is that empirically, the generation interval has been shown to change over time (DOI: 10.1126/science.abc9004), but these changes are not straightforward to include in compartment models, and not accounted for here.

7. Could also be worth discussing other methods used to study geographic spread of influenza and COVID-19 epidemics (e.g. gravity models or network models).

Minor - There are a few editing mistakes (e.g. repeated words, and one sentence in Portuguese) in the supplementary methods.

Reviewer: 2

Comments to the Author(s)

See attached report

===PREPARING YOUR MANUSCRIPT===

===PREPARING YOUR REVISION IN SCHOLARONE===

<https://royalsociety.org/journals/authors/author-guidelines/#data>. You should ensure that

you cite the dataset in your reference list. If you have deposited data etc in the Dryad repository, please include both the 'For publication' link and 'For review' link at this stage.

Author's Response to Decision Letter for (RSOS-210700.R0)

See Appendices B & C.

RSOS-220005.R0

Review form: Reviewer 2

Is the manuscript scientifically sound in its present form?

No

Are the interpretations and conclusions justified by the results?

No

Is the language acceptable?

Yes

Do you have any ethical concerns with this paper?

No

Have you any concerns about statistical analyses in this paper?

No

Recommendation?

Reject

Comments to the Author(s)

The reviewer is saddened that the authors did not fully understand their mistake in the article as explained in the first review. The incidence of infections is determined by the mechanisms of transmission in the model. In contrast, incidence of case reports (data) depends largely on the testing, which is performed on people who resented themselves for a test, these include people

who were infected many days ago who happened to be tested on that day. The incidence of infections and the incidence of case reports are two entirely different concepts in epidemiology. In the authors' responses, the statement "we decided to use the newly registered cases as a proxy to new infections" is scientifically wrong, because the newly reported cases may include people who were infected many days ago, as well as people infected more recently. It should not be used as a proxy for the number of new infections on a particular day, even when the report delays or disease latency are negligible.

While the authors stated that applications in Section 4 is merely a demonstration, such a demonstration based on mixing different concepts will mislead many readers.

Review form: Reviewer 3

Is the manuscript scientifically sound in its present form?

Yes

Are the interpretations and conclusions justified by the results?

No

Is the language acceptable?

Yes

Do you have any ethical concerns with this paper?

No

Have you any concerns about statistical analyses in this paper?

No

Recommendation?

Accept with minor revision (please list in comments)

Comments to the Author(s)

The authors present a very interesting contribution to the literature which provide a useful analytical framework for estimating $R(t)$. The manuscript has already undergone significant revision and I do not have any concerns regarding the mathematical models presented. However, I had difficulty contextualizing the importance of this work in terms of understanding what real-world problems it might solve. Specifically:

1) As with any model fitting exercise, it is imperative that the authors clearly state what parameters are actually being fitted. This is in the supplementary material part 3, but it really should be in the main text. I am particularly concerned by the fact that, as I understand it, the authors are fitting what seems to be a parameter per datapoint (i.e., $\text{Beta}(i,t)$ for each timepoint where data was observed in each metapopulation). I may be misunderstanding the methodology, but this seems like potential overfitting. If I have misunderstood this, it should be clarified. How do these fits account for uncertainty?

2) To that end, I think the manuscript could greatly benefit from comparing estimates from this model with a simpler one. Perhaps the authors could highlight what results would not be obtainable with existing methods. For example, why wouldn't a normal metapopulation SEIR work in this context? It seems to me that the main difference would involve a simpler model having to assume kind of functional form to account for time-varying $\text{Beta}(t)$. In other words, it is

not clear to me why an alternative methodology would NOT be able to produce "estimates of the time series" and "the number of exported cases."

3) Given that this is a paper about estimating time-varying reproductive rate, I am surprised that the final time-varying estimates for each municipality were not shown in a figure. Do the estimates seem sensible? Again, how do they compare to estimates obtained using other methods?

4) If the authors are not accounting for variation in case ascertainment and underreporting over time and space, I would be wary of overinterpreting any specific results from the model fitting portion of this work (e.g., even calling a particular municipality a hub). Indeed, I would argue that properly accounting for uncertainty is a cornerstone of rigorous epidemic modeling.

Decision letter (RSOS-220005.R0)

Dear Mr Jorge,

The Editors assigned to your paper RSOS-220005 "Estimating the effective reproduction number for heterogeneous models using incidence data" have now received comments from reviewers and would like you to revise the paper in accordance with the reviewer comments and any comments from the Editors. Please note this decision does not guarantee eventual acceptance.

Please submit your revised manuscript and required files (see below) no later than 21 days from today's (ie 01-Mar-2022) date. Note: the ScholarOne system will 'lock' if submission of the revision is attempted 21 or more days after the deadline. If you do not think you will be able to meet this deadline please contact the editorial office immediately.

on behalf of Professor Tim Rogers (Associate Editor) and Mark Chaplain (Subject Editor)
 openscience@royalsociety.org

Associate Editor Comments to Author (Professor Tim Rogers):

As well as addressing the other points raised, please be sure to make a substantial revision to the manuscript in order to fully address the strong concerns of reviewer 1 on the estimation of new infections.

Reviewer comments to Author:

Reviewer: 2

Comments to the Author(s)

The reviewer is saddened that the authors did not fully understand their mistake in the article as explained in the first review. The incidence of infections is determined by the mechanisms of transmission in the model. In contrast, incidence of case reports (data) depends largely on the testing, which is performed on people who resented themselves for a test, these include people who were infected many days ago who happened to be tested on that day. The incidence of infections and the incidence of case reports are two entirely different concepts in epidemiology. In the authors' responses, the statement "we decided to use the newly registered cases as a proxy to new infections" is scientifically wrong, because the newly reported cases may include people who were infected many days ago, as well as people infected more recently. It should not be used as a proxy for the number of new infections on a particular day, even when the report delays or disease latency are negligible.

While the authors stated that applications in Section 4 is merely a demonstration, such a demonstration based on mixing different concepts will mislead many readers.

Reviewer: 3

Comments to the Author(s)

The authors present a very interesting contribution to the literature which provide a useful analytical framework for estimating $R(t)$. The manuscript has already undergone significant revision and I do not have any concerns regarding the mathematical models presented. However, I had difficulty contextualizing the importance of this work in terms of understanding what real-world problems it might solve. Specifically:

1) As with any model fitting exercise, it is imperative that the authors clearly state what parameters are actually being fitted. This is in the supplementary material part 3, but it really should be in the main text. I am particularly concerned by the fact that, as I understand it, the authors are fitting what seems to be a parameter per datapoint (i.e., $\text{Beta}(i,t)$ for each timepoint where data was observed in each metapopulation). I may be misunderstanding the methodology, but this seems like potential overfitting. If I have misunderstood this, it should be clarified. How do these fits account for uncertainty?

2) To that end, I think the manuscript could greatly benefit from comparing estimates from this model with a simpler one. Perhaps the authors could highlight what results would not be obtainable with existing methods. For example, why wouldn't a normal metapopulation SEIR work in this context? It seems to me that the main difference would involve a simpler model having to assume kind of functional form to account for time-varying $\text{Beta}(t)$. In other words, it is

not clear to me why an alternative methodology would NOT be able to produce "estimates of the time series" and "the number of exported cases."

3) Given that this is a paper about estimating time-varying reproductive rate, I am surprised that the final time-varying estimates for each municipality were not shown in a figure. Do the estimates seem sensible? Again, how do they compare to estimates obtained using other methods?

4) If the authors are not accounting for variation in case ascertainment and underreporting over time and space, I would be wary of overinterpreting any specific results from the model fitting portion of this work (e.g., even calling a particular municipality a hub). Indeed, I would argue that properly accounting for uncertainty is a cornerstone of rigorous epidemic modeling.

===PREPARING YOUR MANUSCRIPT===

If you have been asked to revise the written English in your submission as a condition of publication, you must do so, and you are expected to provide evidence that you have received language editing support. The journal would prefer that you use a professional language editing service and provide a certificate of editing, but a signed letter from a colleague who is a fluent speaker of English is acceptable. Note the journal has arranged a number of discounts for authors using professional language editing services (<https://royalsociety.org/journals/authors/benefits/language-editing/>).

===PREPARING YOUR REVISION IN SCHOLARONE===

Author's Response to Decision Letter for (RSOS-220005.R0)

See Appendix D.

RSOS-220005.R1

Review form: Reviewer 2

Is the manuscript scientifically sound in its present form?

Yes

Are the interpretations and conclusions justified by the results?

Yes

Is the language acceptable?

Yes

Do you have any ethical concerns with this paper?

No

Have you any concerns about statistical analyses in this paper?

No

Recommendation?

Accept with minor revision (please list in comments)

Comments to the Author(s)

It is commendable that the authors fixed the problem of using the test data to fit the incidence term in the model. Back-calculation method is a standard way to estimate the infected population from testing data. However, the standard back-calculation method assume that the time from infection to diagnosis has a fixed distribution (as the one the authors referred to in the paper). This assumption is reasonable for endemic diseases such as HIV infections. For COVID-19 epidemics, it is very typical that the number of tests performed by public health agencies vary significantly over the course of a wave, with the number of daily test near the peak time doubles that at the beginning and the end of the wave. This means that the distribution is time-dependent. This should be commented on as a limitation of the method in the discussion, so that the reader will be aware.

Decision letter (RSOS-220005.R1)

Dear Mr Jorge

On behalf of the Editors, we are pleased to inform you that your Manuscript RSOS-220005.R1 "Estimating the effective reproduction number for heterogeneous models using incidence data" has been accepted for publication in Royal Society Open Science subject to minor revision in accordance with the referees' reports. Please find the referees' comments along with any feedback from the Editors below my signature.

Please submit your revised manuscript and required files (see below) no later than 7 days from today's (ie 01-Aug-2022) date. Note: the ScholarOne system will 'lock' if submission of the revision is attempted 7 or more days after the deadline. If you do not think you will be able to meet this deadline please contact the editorial office immediately.

on behalf of Professor Tim Rogers (Associate Editor) and Mark Chaplain (Subject Editor)
openscience@royalsociety.org

Reviewer comments to Author:

Reviewer: 2

Comments to the Author(s)

It is commendable that the authors fixed the problem of using the test data to fit the incidence term in the model. Back-calculation method is a standard way to estimate the infected population from testing data. However, the standard back-calculation method assume that the time from infection to diagnosis has a fixed distribution (as the one the authors referred to in the paper). This assumption is reasonable for endemic diseases such as HIV infections. For COVID-19 epidemics, it is very typical that the number of tests performed by public health agencies vary significantly over the course of a wave, with the number of daily test near the peak time doubles that at the beginning and the end of the wave. This means that the distribution is time-dependent. This should be commented on as a limitation of the method in the discussion, so that the reader will be aware.

===PREPARING YOUR MANUSCRIPT===

one version should clearly identify all the changes that have been made (for instance, in coloured highlight, in bold text, or tracked changes);

===PREPARING YOUR REVISION IN SCHOLARONE===

- Ensure that your data access statement meets the requirements at <https://royalsociety.org/journals/authors/author-guidelines/#data>. You should ensure that you cite the dataset in your reference list. If you have deposited data etc in the Dryad repository, please only include the 'For publication' link at this stage. You should remove the 'For review' link.
- If you are requesting an article processing charge waiver, you must select the relevant waiver option (if requesting a discretionary waiver, the form should have been uploaded, see 'File upload' above).
- If you have uploaded any electronic supplementary (ESM) files, please ensure you follow the guidance at <https://royalsociety.org/journals/authors/author-guidelines/#supplementary-material> to include a suitable title and informative caption. An example of appropriate titling and captioning may be found at https://figshare.com/articles/Table_S2_from_Is_there_a_trade-off_between_peak_performance_and_performance_breadth_across_temperatures_for_aerobic_scope_in_teleost_fishes_/3843624.

Author's Response to Decision Letter for (RSOS-220005.R1)

See Appendix E.

Decision letter (RSOS-220005.R2)

Dear Mr Jorge:

I am pleased to inform you that your manuscript entitled "Estimating the effective reproduction number for heterogeneous models using incidence data" is now accepted for publication in Royal Society Open Science.

Please remember to make any data sets or code libraries 'live' prior to publication, and update any links as needed when you receive a proof to check - for instance, from a private 'for review' URL to a publicly accessible 'for publication' URL. It is also good practice to add data sets, code and other digital materials to your reference list.

Royal Society Open Science is a fully open access journal. A payment may be due before your article is published. Our partner Copyright Clearance Center's RightsLink for Scientific Communications will contact the corresponding author about your open access options from the email domain @copyright.com (if you have any queries regarding fees, please see <https://royalsocietypublishing.org/rsos/charges> or contact authorfees@royalsociety.org).

on behalf of Professor Tim Rogers (Associate Editor) and Professor Mark Chaplain (Subject Editor).

Follow Royal Society Publishing on Twitter: @RSocPublishing
Follow Royal Society Publishing on Facebook:
<https://www.facebook.com/RoyalSocietyPublishing/>
Read Royal Society Publishing's blog:
<https://royalsociety.org/blog/blogsearchpage/?category=Publishing>

Appendix A

Review of RSOS-210700, “Estimating the effective reproduction number for heterogeneous models using incidence data”

The authors introduced a theoretical method to derive the effective reproduction number for a general class of epidemic models with infection age structure, which include many complex and heterogeneous models. One of the advantages of the method is that it can overcome the difficulty of lack of data to inform serial intervals needed for the estimation of the effective reproduction number. Another highlight of the paper is to apply the method to estimate effective production numbers for the COVID-19 dynamics in Brazil, using real world incidence data.

The mathematical derivations are sound. But the authors’ treatment of linkage to data is careless, problematic, and erroneous, and it seriously diminishes the significance of the results in the paper.

To explain the where the authors were not careful about introducing the data into their models, we need to understand what is the real-world incidence data, and how is it related to transmission models. The real-world incidence data (used in the paper) is a time series of daily reported number of positive COVID-19 cases in a jurisdiction (Brazil). This is called in the epidemiological or public health literature as “incidence of cases”, and should not be confused with the “incidence of infections” in a transmission model, which means the number of new infections.

How does a new infection become a new case and get recorded in the public health database? First, the newly infected person has to have a reason to seek COVID-19 test, typically after feeling some of symptoms a few days after infection or having recently contacted people who tested positive, then the test has to show positivity, and then the positive test needs to be recorded and reported to central database, often with days of delay. Positive cases are therefore a subset of infected individuals with various infection ages. In reality, it is highly unlikely for a COVID-19 test to catch an infected individual on the same day of the infection, namely at infection age 0.

The connection to data is dealt with in Section 2-(d), on page 7. In (2.23), parameters ρ_i is introduced to describe “a higher or lower notification”. This is supposed to be the fraction of infected people who are tested positive on a particular day (the typical length of Δt). However, the parameter ρ_i is multiplied to the cumulation (integral over Δt of new infections \mathcal{F}_i). These people have an infection age 0, as indicated in the original model (2.1)-(2.2). This mistake is a result of mismatching data with model outcomes.

Another way to see why (a fraction of) new infections \mathcal{F}_i should not be identified with new cases data is that people who tested positive for COVID-19 typically are immediately isolated or hospitalized if the symptom is severe, so it would be unlikely for them to infect others. Since the ρ_i is assumed to be 1 (100%) for infected people who are symptomatic (in the supplementary material, page 14, two lines below (76)), the identification used in

(2.23) means that symptomatically infected will be isolated upon infection and will not have any chance to infect anyone. This is certainly not what the authors intended in the model.

Including the parameters ρ_i in the incidence term \mathcal{F}_i in the model as in (2.23) also presents a theoretical problem for the authors. In formula (2.25) on page 8 of the manuscript, fractions ρ_i/ρ_j are used. As assumed in the supplementary materials, page 14, three lines below (76), “only symptomatic individuals report their infections”, which means that asymptomatic individuals will not report their infections, and thus $\rho_3 = 0$ for the I_2 group, and one of the terms ρ_i/ρ_j in the denominator of \mathcal{R}^T is infinitely large, and thus \mathcal{R}^T will be infinitely small, just because asymptotically infected people not getting tested. This is very counter intuitive, and needs to be addressed.

On page 8 of the main manuscript, in formula (3.1), parameters ρ_i disappeared from the formula of R_{ij} . This can be very misleading, because some readers might use this formula to compute the effective reproduction number irrespective of the values of ρ_i .

If the case data should not be identified with the incidence of infections, how should one incorporate the case data into the model? The correct way is to introduce a case compartment C_i for each X_i , and a flow $\rho_i x_i$ from X_i to C_i , where ρ_i indicates the fraction of infected in compartment X_i get tested positive. The daily new case data can be identified with $\rho_i x_i$. This way, COVID-19 testing plays the role of identify and isolate infected individuals and prevent them from further infecting others. If one has data to inform the values of ρ_i , then the data can be used to identify x_i and estimate other parameters in the model.

Appendix B

Dear Professor Tim Rogers (Associate Editor) and Mark Chaplain (Subject Editor) Royal Society Open Science Editorial Office

Thank you for providing a review of our manuscript RSOS-210700 "Estimating the effective reproduction number for heterogeneous models using incidence data". We also thank both reviewers for their comments on the reports. In the revised version, which we resubmit to RSOS, we have considered the criticism and comments in the reports to modify the original manuscript.

The text was entirely reviewed and edited to eliminate grammatical and stylistic flaws, in such a way to better convey our points of view on the subject. We also enlarged the text to discuss some issues raised by the reviewers. Thus we now provide an updated discussion of recent advances in the evaluation of the effective reproduction number and age distributions, as well as available methods to improve the quality of reported data on the number of daily reported cases by health authorities. Accordingly, new bibliographic items suggested by both reviewers were added to the reference list. All changes in the original text are printed in red.

As required, all questions and comments raised in the report have also been addressed in our point-by-point answer included in the resubmission material. Here again, for the sake of better identification, our answers and comments to the reviewers are highlighted in red.

At this point, it is important to emphasize our understanding that the main result of this work is to present a general framework to investigate and evaluate the effective reproduction number and age distribution for heterogeneous models. As acknowledged by reviewer 1, such a broad approach to this task seems to be missing, and our results may become a relevant contribution to the field. In spite of that, we do think the examples we provided in section 3 play an important role to indicate how the analytical expressions can be derived for simple heterogeneous models in a clear and transparent way. This is an important step to encourage potential readers to use our method to treat their own realistic and more complex models. Along the same way, the examples in section 4 are primarily intended to exemplify how to use our approach together with the available data to obtain two important results for the effective reproduction number: i) the mutual influence of pairs of meta-populations on giving rise to new cases; ii) the general effect caused in the whole ensemble of meta-populations due to their interaction. Therefore, as in previous papers published by us or by other authors, we actually did not get involved in providing higher quality estimates to the actual number of new infections and decided to use the officially reported number of daily cases. To our opinion, if one gets involved with this issue it would demand great efforts and produce enough results to justify writing another work. We understand that referee 2 and the editor have correctly addressed the possible lack of accuracy of our results on COVID-19 in section 4. However, one may consider that limitation as an effect only due to the used data, which does not affect the validity of the proposed method.

Finally, in order to make clear the scope of our manuscript, we ask you to consider the possibility of changing its title to "Estimating the effective reproduction number for heterogeneous compartment models". By doing this, we would not set the expectation of the

reader to a detailed treatment of incidence data of either COVID-19 or any other disease, which is not the main focus of the work.

We hope the editor takes into due consideration our arguments regarding the use of the same data set as in the first submitted version, and that the substantial changes we carried out in the manuscript have made it suitable for publication in RSOS.

Yours sincerely

D. C. P. Jorge, on behalf of the authors.

Appendix C

Reviewer 1 comments with replies

In this manuscript, the authors present a general approach to deriving the generation interval distribution and the time-varying reproductive number within an ODE model that includes host heterogeneity (e.g. heterogeneity in age, space, risk group). This builds on previous work describing how to calculate the basic (but not the time-varying) reproductive number in models of simple or heterogeneous populations. It also builds on previous work describing how to derive the generation interval within an SEIR-type model (without heterogeneity) (<https://doi.org/10.1137/18M1186411>).

From what I can assess, the work seems sound, and useful. Differential equations are not my area of expertise, and I was not able to assess every detail of the derivations presented here. However, throughout the derivation I can follow the general logic, and can draw mathematical parallels between the equations/approach presented here, and methods I've used before when working with simpler SEIR-type, or empirical models for estimating $R(t)$. The general logic and mathematics of the derivations shown in section 2 seem reasonable, and they simplify in ways consistent with previously published work.

Our Reply:

We are very thankful for the general comment about our work and the careful report of the reviewer, catching the essential features of our work. Before addressing the specific comments and suggestions in the report, we would like to inform that, in order to better convey the scope of our manuscript, we have asked the editor to consider the possibility of changing its title to “Estimating the effective reproduction number for heterogeneous compartment models”. By doing this, we believe to adequately highlight our aim of presenting a general method to derive the generation interval distribution and reproduction number from compartment heterogeneous models. In spite of the suggestion to change the title and in order to illustrate the method, we have also kept the evaluations of the reproduction number to the COVID-19 data in Brazil, as in the original submission.

Comment 1: *The main weakness of the submission is that, at several key points in the text, the writing is difficult to understand. I think it's unfair to researchers in non-English speaking countries that publications are almost always written in English. But I encourage the authors to seek editorial help from a native English speaker if possible. Overall, the scientific content seems good, but the impact of this paper will be greater if*

Reply: We thank the reviewer for bringing this issue to our attention. We have carried out thorough proofreading of the whole text, and hope to have eliminated all grammatical and stylistic flaws that may make it difficult to accurately understand our ideas. We also emphasize that the Discussion has been substantially altered to meet the comments of another reviewer regarding the impact of the quality of the data on the results of the model. In the new version, the changes are written in red.

the ideas are easier for readers to understand (more specifics are discussed below).	
Comment 2 (First part): The significance of the work, and its relationship to existing methods, is not explained very clearly in the abstract/introduction. It would be helpful to draw a clearer distinction between empirical approaches to estimating $R(t)$ and the generation interval, and inference of $R(t)$ within compartment models, where the generation interval must be derived from the equations of the model. Empirical methods are not really the focus of this manuscript – these methods are already well-established and widely used (pre-2020 R_t methods reviewed in - https://doi.org/10.1371/journal.pcbi.1008409, generation interval estimation - https://doi.org/10.1073/pnas.2011548118, newer R_t methods - https://doi.org/10.1101/2020.09.14.20194589, see also the recently developed R packages EpiNow2, Epifilter, Epidemia, and others).	Reply: We thank the reviewer for this comment. We have rewritten most of the Introduction. Now, the third paragraph presents a discussion on different strategies to evaluate $R(t)$, namely the empirical methods based on available data and statistical basis, and those methods relying on compartmental models. New literature items suggested by the reviewer have been included. The Introduction also advances the advantage of the adopted approach, which consists in considering both the dynamics assigned by the model setup and the available data. This allows rescuing, for instance, the important influence of unregistered cases as the asymptomatic patients in the dynamics of COVID-19, as detailed discussed in Sections 3 and 4.
Comment 2 (Second part): From the abstract -- “However, there are systems, especially highly heterogeneous ones, in which there is a lack of data and an adequate methodology to obtain $g(\tau)$.” As currently written, this claim seems overly broad, as it’s not clear if the manuscript is trying to argue that: a. There are no adequate methods to estimate $g(\tau)$ empirically from surveillance data, in which heterogeneities between hosts exist inevitably. (this statement is not true, see -- - https://doi.org/10.1073/pnas.2011548118; https://doi.org/10.1016/j.epidem.2018.12.002), or	Reply: The reviewer is correct in identifying the misleading statement in the Abstract. Indeed, our intention was to emphasize only the aspect included in part (b) of the reviewer’s partition of the sentence. This has now been clarified in the new version of both Abstract and in the third paragraph of the Introduction. In fact, to the best of our knowledge, there wasn’t any general method to estimate the effective reproduction number and generation interval distribution for heterogeneous compartment models based on the next-generation method used to estimate the basic reproduction number R_0. In this manuscript, we present a general method that leads to matrices for the reproduction number and generation interval distribution. We also show that, for some restrictive conditions, we can further reduce the final result to one single expression for the

b. Methods to derive the generation interval distribution from a compartment model that includes host heterogeneity have not been established (to my knowledge, this is true, and I think it's what the authors are trying to say).	reproduction number and for $g(\tau)$.
Comment 2 (Third part): It would be helpful to clarify what the authors mean by "highly heterogeneous systems". For example, making it clear that the authors are referring to models that include heterogeneities such as age, geographic or risk structure in the host population would be helpful.	Reply: By highly heterogeneous models we refer to any system in which heterogeneities, either in the host population or in the expression of the disease, are very impactful in the disease transmission process. The reviewer is correct in including age, geographic, or risk structure in the host population as examples of heterogeneities that can be dealt with by our approach. All of them increase the dimension of the system of non-linear equations generating, for instance, different classes of susceptible or infectious individuals. Nevertheless, we understand that the term is very ambiguous and we have removed it from the text and provided a more careful explanation of the argument, whenever necessary.
Comment 2 (Fifth part): Clarify that this method can produce separate $R_{ij}(t)$ estimates for transmission within and between model sub-compartments. (This is innovative!)	Reply: We appreciate the referee's comment. The estimation of distinct elements $R_{ij}(t)$ may be very useful for the understanding of the spreading dynamics. For instance, in a geographic meta-population model, we called the attention that R_{ij} links the generated infected cases in compartment "i" due to the infectious cases in compartment "j". Additionally, they allow us to evaluate the "infection contribution" of each county to each other as well. We discuss this point in the Introduction and return to it in the Discussion.
Comment 3: This may be obvious to the authors, but why isn't it correct/sufficient to estimate R_0 using the next generation matrix method, and then obtain $R(t)$ using the equation: $R(t) = R_0 \cdot S(t)$? While not necessary for publication, it would be quite helpful for readers and users of these methods to see how badly this easier	Reply: That question is relevant and helps us to highlight another advantage of using the proposed method. The result $R(t) = R_0 \cdot S(t)$ is valid only for simple models such as SIR and SEIR. The arguments leading to it fail immediately when one considers more than one susceptible compartment or parameters that change through time. Though one can always define one single R_0, that is not the case for the $R(t)$, as we also show in the

approach performs relative to the exact solution presented here.	manuscript. We included a sentence at the end of the first paragraph of the Discussion section to call the attention of the reader to the limit of validity of this expression.
Comment 4: Sometimes the notation is not explained clearly, or used inconsistently, which makes it difficult to follow the mathematical arguments. E.g. from page 4 -  • Is it true (25) that x can really be interpreted as the “infection age distribution?” In the previous sentence, the authors state that x is a vector whose entries represent individuals infected at the same time (i.e. of the same infection age), but from different population sub compartments. • line 32 - the meaning of the V and F matrices could be explained more clearly. Readers already familiar with the next generation method should be able to follow, but re-wording would be helpful! • Maybe this is a mathematical convention that I’m not familiar with, but at various points I found it confusing that the x notation switched between capital and lowercase letters. It seems like X is used to indicate totals across all tau values, but an explanation could be helpful. 	Reply : We thank the reviewer for this comment. The notation was indeed confusing throughout the text. We have carefully worked on section 2 in order to clarify those points. Please find below our answers to the specific points highlighted in the comment:  • x is defined as a vector that encompasses the infection age distributions. Each element of this vector gives the infection age distribution of a given compartment. This is now as clearly indicated in the text. • Thanks for this suggestion. In the new version, we tried to explain the meaning of the V and F vectors in the third paragraph of section (2a) of the manuscript. • Yes, the capital letter (X) indicates the integral over all tau values of the age generation distribution of the total number of individuals of a given compartment, which is indicated by the lower case letter (x). We explicitly indicate it in the second paragraph of section 2b in the new version of the manuscript.
Comment 5: It’s interesting that $g_{ij}(\tau)=g(\tau)$ in all the examples presented in the supplement and main text. Can the authors make any general statements about when this is or is not true?	Reply: This is a very good observation. This is valid for both used meta-population models because we assumed that the recovery process is the same in all meta-populations. In the Supplementary Material, there are also examples in which this is not the case. In the new text, we briefly highlight this in the paragraph after Eq. 3.1.
Comment 6: A point worth discussing is that empirically, the generation interval has been shown to change over time (DOI: 10.1126/science.abc9004), but these changes are not straightforward to include in compartment models, and not accounted for here.	Reply: Actually, the generation interval distribution obtained by our method can change over time. If one constructs a compartmental model with parameters that change in time, the generation interval distribution may also change in time. An easy way to see it is to introduce time dependency on the recovery parameter of the SIR model (γ). We highlight this point in the text after Eq. 2.14 and include the article the reviewer suggested to the references.

Comment 7: Could also be worth discussing other methods used to study geographic spread of influenza and COVID-19 epidemics (e.g. gravity models or network models).	Reply: Thank you for the suggestion, we address this shortly in the discussion.
Minor: There are a few editing mistakes (e.g. repeated words, and one sentence in Portuguese) in the supplementary methods.	Reply: We thank the reviewer for calling our attention to the misprints, which have been fixed

Reviewer 2 comments with replies

Review of RSOS-210700, “Estimating the effective reproduction number for heterogeneous models using incidence data”. The authors introduced a theoretical method to derive the effective reproduction number for a general class of epidemic models with infection age structure, which include many complex and heterogeneous models. One of the advantages of the method is that it can overcome the difficulty of lack of data to inform serial intervals needed for the estimation of the effective reproduction number. Another highlight of the paper is to apply the method to estimate effective production numbers for the COVID-19 dynamics in Brazil, using real world incidence data.

Our Reply:

We appreciate the relevant criticisms of our work and the spent time by the reviewer to perform a careful report.

The criticism on questions related to data quality, which are certainly important but which we have not taken as the main focus of our work, motivated us to ask the editor to consider the possibility of changing its title to “Estimating the effective reproduction number for heterogeneous compartment models”.

By doing this, we would not set the expectation of the reader to a detailed treatment of incidence data of either COVID-19 or any other disease, which is not the main focus of the work. The objective in presenting specific results based on COVID-19 data from Brazil, which have been kept in the revised manuscript, is to illustrate how the method is used to derive the generation interval distribution and reproduction number from compartment heterogeneous models.

Comment 1 : *The mathematical derivations are sound. But the authors’ treatment of linkage to data is careless, problematic, and erroneous, and it seriously diminishes the significance of the results in the paper. To explain where the authors were not careful about introducing the data into their models, we need to understand what is the real-world incidence data, and how is it related to transmission models. The real-world incidence data (used in the paper) is a time series of daily reported number of positive COVID-19 cases in a jurisdiction (Brazil). This is called in the epidemiological or public health literature as “incidence of cases”, and should not confused with the “incidence of infections” in a transmission model, which means the number of new infections.*

Reply: We thank the reviewer for this clarifying comment, which is valid actually for any work related to using reported data on modeling of actual epidemic processes.

For sure, the actual reported data for any epidemic process faces intrinsic problems related to the delay between infection and case notification as well as to sub-notification associated with low testing. The methodology developed in the manuscript and applied in section 4 makes no assumption on the quality of the time series for the new infections. Because of that, it does not need to be changed in order to introduce more reliable data for the new cases. Although being aware of recent progress in the literature that focuses on providing better estimates for the new infections series out of the registered data, we only briefly mentioned problems arising from the limited quality of data in the

How does a new infection become a new case and get recorded in the public health database? First, the newly infected person has to have a reason to seek COVID-19 test, typically after feeling some of symptoms a few days after infection or having recently contacted people who tested positive, then the test has to show positivity, and then the positive test needs to be recorded and reported to central database, often with days of delay. Positive cases are therefore a subset of infected individuals with various infection ages. In reality, it is highly unlikely for a COVID-19 test to catch an infected individual on the same day of the infection, namely at infection age 0.

last part of the Discussion section.

Taking into account the reviewer's comment, in sub-sections (2a)-(2c) we now clarify that the presented theoretical analysis only deals with the new infections and not cases, and in the sub-section 2d, we include a more consistent discussion regarding the problem of data quality and indicated some available methods to sidestep some of its undesired effects in the modeling results. The sub/super-notification factors were also removed from the manuscript since they are related to an independent treatment of the data. Nevertheless, we understand that the notification issues may be dealt with independently from the developed methodology and, once the new infections time series are estimated, applying the method presented in the manuscript is straightforward.

We emphasize that the application, in section 4, is intended to serve as an illustrative example to show how to apply the method. In line with this, we decided to use the newly registered cases as a proxy to new infections, even if it may be a crude approximation. Meanwhile, for the metapopulation SIR-type model, it is associated with the infected compartment for each county, for the metapopulation SEIR model, it is related to the symptomatic infected compartment. Here, the new cases are associated with the fraction of symptomatic cases, p . The changes were made both in section (2d) of the manuscript and in the Supplementary Material 3. Nevertheless, we mention again the impact of the limited quality in the database subsection of section 4, and in the last paragraphs of the Discussion, we highlight again the limitation of our results caused by the limited quality of the used data set.

Comment 2: *Another way to see why (a fraction of) new infections F_i should not be identified with new cases data is that people who tested positive for COVID-19 typically*

Reply: The reviewer's remark is correct and, indeed, this is not what we intended in the model. The self-quarantine possibility is a feature that can be introduced in compartmental models and, in such case, the developed method will capture its effect and

are immediately isolated or hospitalized if the symptom is severe, so it would be unlikely for them to infect others. Since the ρ_i is assumed to be 1 (100%) for infected people who are symptomatic (in the supplementary material, page 14, two lines below (76)), the identification used in (2.23) means that symptomatically infected will be isolated upon infection and will not have any chance to infect anyone. This is certainly not what the authors intended in the model.	translate it into the reproduction number and generation interval distribution. However, in the scope of the application sections, this is not what we are intending to do. As clarified in the last comments, we are not looking for a rigorous data analysis. Our purpose with the application section is mainly illustrative; concerning the metapopulation models, we highlight the flux of infected people between the counties. The models and the data treatment are very simple and we had acknowledged their limitations in the Discussion.
Comment 2 (First part): Including the parameters ρ_i in the incidence term F_i in the model as in (2.23) also presents a theoretical problem for the authors. In formula (2.25) on page 8 of the manuscript, fractions ρ_i/ρ_j are used. As assumed in the supplementary materials, page 14, three lines below (76), “only symptomatic individuals report their infections”, which means that asymptomatic individuals will not report their infections, and thus $\rho_3 = 0$ for the I2 group, and one of the terms ρ_i/ρ_j in the denominator of RT is infinitely large, and thus RT will be infinitely small, just because asymptotically infected people not getting tested. This is very counter intuitive, and needs to be addressed.	Reply: We thank the referee for calling our attention to this confusing aspect. Indeed, this problem arises when we consider the trivial case, in which the notification is zero. In this case, the expression for the $B_i(t)$ should read $B_i(t)=0$. However, to avoid unnecessary misunderstanding we have abandoned the sub/super-notification factors as, in fact, it is not an essential aspect of the method. Therefore, this problem is no longer present.
Comment 3: On page 8 of the main manuscript, in formula (3.1), parameters ρ_i disappeared from the formula of R_{ij}. This can be very misleading, because some readers might use this formula to compute the effective reproduction number irrespective of the values of ρ_i	Reply: As already addressed in the previous reply, the parameter ρ_i only appeared on the formulas for estimation using the new infections series, but it no longer appears in the new version. On the other hand, equations (3.1) are the analytical expression for the $R(t)$ and $g(\tau)$ for the SIR meta-population model. These expressions should only depend on the parameters of the model. They are correct as they are presented, and its derivation can be found in Supplementary Material 1.
Comment 4: If the case data should not be identified with the incidence of infections, how	Reply : We thank the referee for the suggestion, although we think that including

should one incorporate the case data into the model? The correct way is to introduce a case compartment C_i for each X_i , and a flow $p_i x_i$ from X_i to C_i , where p_i indicates the fraction of infected in compartment X_i get tested positive. The daily new case data can be identified with $p_i x_i$. This way, COVID-19 testing plays the role of identify and isolate infected individuals and prevent them from further infecting others. If one has data to inform the values of p_i , then the data can be used to identify x_i and estimate other parameters in the model.

so many details in the model deviates from the purpose of section 4. Indeed, its main objective is to provide a simple application of the developed method to a problem of current relevance. Several relevant contributions in the literature on Covid-19 modeling, including some by us, have used the officially reported data, in spite of their deficiencies (see for instance references [14], [24], [26], [28], [48], [39], [42] and [43]).

Besides that, including new compartments to keep track of the case registration would indeed be a way of taking into account the delays between infection and case notification which may be different for each county.. However, we remind an intrinsic difficulty, as modeling how the flow would occur between the X_i and C_i is somewhat of arbitrary choice, and reflects the dynamics of notification of each specific disease. We remind again that we introduced a general method and we don't want to restrict the results to some specific scenarios.

There are already several methods for estimating the new infections from the new cases, each one with its own assumptions and methodology. Therefore we want to give the reader the flexibility to choose the method that they think is more suitable for their specific system. As we have already highlighted in the previous reply, the problems regarding the delay between infection and notification have been conveniently addressed in sub-sections (2d) and (4a), as well as in the Discussion. Additionally, we present references [33], [34], and [35] for the methods of nowcasting and back projection.

Appendix D

Reviewer: 2

Comments to the Author(s)

The reviewer is saddened that the authors did not fully understand their mistake in the article as explained in the first review. The incidence of infections is determined by the mechanisms of transmission in the model. In contrast, incidence of case reports (data) depends largely on the testing, which is performed on people who resented themselves for a test, these include people who were infected many days ago who happened to be tested on that day. The incidence of infections and the incidence of case reports are two entirely different concepts in epidemiology. In the authors' responses, the statement "we decided to use the newly registered cases as a proxy to new infections" is scientifically wrong, because the newly reported cases may include people who were infected many days ago, as well as people infected more recently. It should not be used as a proxy for the number of new infections on a particular day, even when the report delays or disease latency are negligible.

While the authors stated that applications in Section 4 is merely a demonstration, such a demonstration based on mixing different concepts will mislead many readers.

Our Reply:

In this version, we reviewed our decision, following the reviewer's suggestion about estimating the incidence time series. Although we had fully understood the arguments in the first report of Reviewer 2 concerning the differences between reported and infected cases, we kept working with the reported data in the second version, supposing that our arguments were clear enough to convince the reviewer and the readers. Indeed, we introduced a discussion in the text concerning the knowledge of the differences between the two data sets, and the fact that the main focus of our application sections stays on the derivation of new analytical expressions to account for the influence of exogenous contacts in the evaluation of $R(t)$ for a meta-population model.

In this third version, we perform a more accurate analysis to obtain the infection data based on a back-projecting method, according to the recommendation of the reviewer. For sure it is positive to present the new results, exhibited in Figures 1, 2, 3, and 4, although they do not differ appreciably from those based on the reported cases, as can be seen by comparing the new and old versions of Figure 2. Back-projection was carried out using the function `BackprojNP` from the `R-packet Surveillance` available at <https://rdr.io/github/jimhester/surveillance/>. Its brief description was added in the second paragraph of the database subsection of the manuscript. There, we present the basic assumptions for applying the back-projection approach to that problem.

To construct the distribution used for the back-projection, we consider the mean time interval between symptom onset and testing in the Southeast region of Brazil, where the state of Rio de Janeiro is located, to be almost the same as that in the period of the epidemics from March to August obtained from new reference [41], which was added to the bibliographic list. It is relevant to consider specific parameter values for the analysed location since they may vary for different regions in a large and diverse country such as Brazil [41], and for different phases of the epidemics.

Since we understand this was the unique criticism raised by the reviewer against the publication in RSOS journal, we hope that our analyses and presented results clarify him/her about that point. Finally, we thank the reviewer for elucidating that point which improves the precision of the generated results when our methodology is applied to actual epidemics data.

Reviewer: 3

Comments to the Author(s)

The authors present a very interesting contribution to the literature which provide a useful analytical framework for estimating $R(t)$. The manuscript has already undergone significant revision and I do not have any concerns regarding the mathematical models presented. However, I had difficulty contextualizing the importance of this work in terms of understanding what real-world problems it might solve. Specifically:

Our Reply:

We thank the reviewer for the valuable comments in regard to our work, all of which have been considered in the new version of the manuscript. Please find below our replies to the specific comments raised in the report. Also note that, due to comments from reviewer 2 with respect to the use of incidence data instead of reported data in our examples, we carried out a back-projection procedure to estimate the incidence data. As a consequence, we inserted in the text a brief explanation of the used method, which also lead to slightly different results shown in Figures 1, 2, 3, and 4.

Comment 1 : *As with any model fitting exercise, it is imperative that the authors clearly state what parameters are actually being fitted. This is in the supplementary material part 3, but it really should be in the main text. I am particularly concerned by the fact that, as I understand it, the authors are fitting what seems to be a parameter per datapoint (i.e., $\beta(i,t)$ for each timepoint where data was observed in each metapopulation). I may be misunderstanding the methodology, but this seems like potential overfitting. If I have misunderstood this, it should be clarified. How do these fits account for uncertainty?*

Reply: We agree with the reviewer's remark on the necessity of transmitting a clear picture of the procedure. In fact, there is no fitting being done in our work. Estimating the reproduction number from the renewal equation is an iterative process where we use the past data to inform what is the reproduction number that allows us to obtain the next data point. The renewal equation is close to a Green function that propagates an initial condition using the deterministic dynamics of a system. Thus, at each point in time, our method estimates its position on the phase space using the data and gives the reproduction numbers, and thus the values of β , that allow the system to move to the next point. To do this, we solve a linear system at each step. There is no minimization being performed with the data, only the iterative process, already widely used in literature, which we extend for an arbitrary system.

In fact, as discussed in Ref[1,8,25,27], the renewal equation based evaluation of $R(t)$ assumes that transmission rates can change in time. In the new version, we provided, before Eq. (3.1) in the main text, a discussion on the reasons why it is defensible and necessary to assume the daily variation of

	$\beta_j(t)$ and $\phi_{ij}(t)$ for the purpose of evaluating $R(t)$. We also emphasize that, in previous publications of the authors, a similar $R(t)$ formulation was able to evidence the changes in β due to the mitigation policies for Covid-19. Our formulation, generalized here, was able to agree nicely with a compartmental model that assumed a functional form in β for its change in time, Ref[25,27].
Comment 2: To that end, I think the manuscript could greatly benefit from comparing estimates from this model with a simpler one. Perhaps the authors could highlight what results would not be obtainable with existing methods. For example, why wouldn't a normal metapopulation SEIR work in this context? It seems to me that the main difference would involve a simpler model having to assume kind of functional form to account for time-varying $\beta(t)$. In other words, it is not clear to me why an alternative methodology would NOT be able to produce "estimates of the time series" and "the number of exported cases."	Reply: This answer is based on the assumption that the reviewer is suggesting to evaluate $R(t)$ based on analytical expressions in terms of the parameters of a metapopulation SEIR model, given a single fit of the data. If this is indeed the case, we must use Eqs. 3.2 and 3.3, expressed in terms of the model parameters and λ_{ij}, whereby we must consider that these coefficients are also expressed in terms of model parameters. The difference between the two approaches relies only on the fact that one should consider just a time-independent value of β_i, which can be evaluated by minimizing the error between a small set of data points and model evolutions and extending this value for a large time interval. Eventually, new values of β_i could be estimated to accurately reproduce changes in the model evolution due to some actual change affecting the spread of the epidemic, like new variants or new habits regarding social distancing and mask usage. By doing this, the evolution of $R(t)$ would be replaced by smooth curves within large intervals, with a slow decay in time, related only to the reduction of the number of the susceptible population. Eventual stepwise changes between these plateaux emerge only as a consequence of changes in the values of β. In this approach, ϕ_{ij} is also assumed to be constant, but their values might also change with time when one observes changes in the flux of commuting population. This formulation is possible, given that one can fit the model to the multiple municipalities' data, which should be very costly. As discussed before, our work does not rely on fittings, however, previous works of ours

	have compared the results of the $R(t)$ using the method generalized in this work and the result of a fitted model Ref[25,27]. We do not affirm that estimations of $R(t)$ can not be obtained by other approaches, we simply clarified what ours does. Indeed, both methodologies indicated above are able to produce results for $R(t)$, although the obtained values may differ qualitatively from each other. Our focus in this work was to generalize a method that is already used in the literature. The models chosen in the application sections, both the SIR and SEIR models, are just examples and one may, with our method, use other models alongside this or any other data set. We argue that our method has the advantage of expressing $R(t)$ as a response of the system to the data due to the transmission dynamics, detecting short-time variations of the data Ref[25,27]. The method may be applied to simple models or to more complex ones with heterogeneities. Since it depends on linear systems for the iterations, it scales much slower as compared to the minimization of a large number of parameters required for meta-population models.
Comment 3: Given that this is a paper about estimating time-varying reproductive rate, I am surprised that the final time-varying estimates for each municipality were not shown in a figure. Do the estimates seem sensible? Again, how do they compare to estimates obtained using other methods?	Reply: The full series of results would be impractical to display on the main paper, once there are 121 time series of the mutual contributions to the reproduction number in our dataset consisting of 11 municipalities. In any case, we now present the results for the reproduction numbers of São Gonçalo and São João do Meriti in Figure 4 (c and d) which may illustrate examples of the time series of $R(t)$ of a pair of cities It is not clear to us what the reviewer means when referring to “other methods”. We have already commented before the evaluation of $R(t)$ in regards to model-based parameter values, relying on fitting. Another possible method would be a data-based only evaluation of $R(t)$, without considering any modeling. As we comment in the introduction, those methods rely on constructing a generation interval distribution through a system-specific epidemic data set. We do not have the available data to do such estimation and it is not our goal with this work. Our aim

	is to highlight the generalization of a method that takes into account both data and modeling to estimate $R(t)$.
Comment 4: If the authors are not accounting for variation in case ascertainment and underreporting over time and space, I would be wary of overinterpreting any specific results from the model fitting portion of this work (e.g., even calling a particular municipality a hub). Indeed, I would argue that properly accounting for uncertainty is a cornerstone of rigorous epidemic modeling.	Reply : Once again we would like to emphasize that there is no fitting in this work. Our method is derived from deterministic models and, as a consequence, provides deterministic results. The uncertainty regarding the results of the method comes from the uncertainty of the data and the parameters used. In relation to the parameters, the uncertainty arises from the impossibility of knowing their exact values. In effect, all parameter values in Table 1 of Supplementary Material 3 are estimations obtained in the current literature on the SARSCov2 pandemic. In a previous publication by some of us (Ref. 25) we carried out a study on the parameter sensibility for a more complex model than the SEIIR version studied here, we do not consider it necessary to reproduce here the same study for an even simpler model. In relation to the data, one might access its uncertainties in many ways. In particular, it is possible to generate samples of the time series using bootstrap methods and, with those samples, apply the method and estimate the uncertainties. Since the fact that the application is not the main issue of our manuscript, only being used to illustrate the usage of the method, we do not carry out those analyses. As a matter of fact, no study can cover all aspects of a given subject so that, as usual, we prefer to indicate the limitations of our work in the discussion section.

Appendix E

Dear Editor,

We are very pleased to know that our work “Estimating the effective reproduction number for heterogeneous models using incidence data” was accepted for publication in the Journal of the Royal Society Open Science, subject to minor revision. In this new version, we address the reviewer's comment and introduce a few lines in the fourth paragraph of the discussion.

We thank the editors and reviewers for acknowledging our contribution to the field.

Yours Sincerely,

Daniel C. P. Jorge, on behalf of all authors.